# Limited generalizability of multivariate brain-based dimensions of child psychiatric symptoms
Bing Xu [1,2], Lorenza Dall'Aglio [1,2], John Flournoy[3], Gerda Bortsova[4], Brenden Tervo-Clemmens [5], Paul Collins[6], Marleen de Bruijne [4,7], Monica Luciana[6], Andre Marquand[8,9], Hao Wang [10], Henning Tiemeier [1,11] ✉ & Ryan L. Muetzel [1,12]

Multivariate machine learning techniques are a promising set of tools for identifying complex brain-behavior associations. However, failure to replicate results from these methods across samples has hampered their clinical relevance. Here we aimed to delineate dimensions of brain functional connectivity that are associated with child psychiatric symptoms in two large and independent cohorts: the Adolescent Brain Cognitive Development (ABCD) Study and the Generation R Study (total $n = 6935$). Using sparse canonical correlations analysis, we identified two brain-behavior dimensions in ABCD: attention problems and aggression/rule-breaking behaviors. Importantly, out-of-sample generalizability of these dimensions was consistently observed in ABCD, suggesting robust multivariate brain-behavior associations. Despite this, out-of-study generalizability in Generation R was limited. These results highlight that the degrees of generalizability can vary depending on the external validation methods employed as well as the datasets used, emphasizing that biomarkers will remain elusive until models generalize better in true external settings.

Psychiatric neuroimaging has sought to illuminate the neurobiological underpinnings of psychiatric disorders over the past few decades, providing a unique opportunity to study child and adolescent neurodevelopment as a key risk window for the emergence of mental health problems[1]. Brain-behavior association studies represent a promising approach to explore individual brain variability that predicts behavioral phenotypes[2–4]. To date, however, clinical translation of brain-behavior predictions has been immensely challenging: rigorously validated and generalizable neurobiological biomarkers that are able to guide clinical practice remain elusive[5–9]. Several features of the literature may account for this difficulty, such as insufficient statistical power (e.g., limited sample size), substantial variability across methodologies, and a heavy reliance on univariate analysis techniques that could fail to map the likely multidimensional neural bases of

psychiatric disorders[8,10,11]. Inherent heterogeneity and high comorbidity of psychiatric disorders exacerbate the problem, rendering it difficult to isolate the most relevant neural features of interest. This is especially the case for children and adolescents who usually present less clearly defined psychopathology and heterotypic continuity of symptoms and phenotypes[12].

Multivariate analyses within machine learning frameworks may permit the detection of associations that have proved to be elusive[4,13]. Several aspects of multivariate techniques make them an appealing choice for studying psychopathology through neuroimaging. First, multivariate methods are less hampered by the small effect sizes that univariate analysis of psychiatric neuroimaging studies typically observe[8,10,11], resulting in greater statistical power and the potential for better reproducibility of brain-behavior associations[4]. Second, multivariate methods with a data-driven

[1]Department of Child and Adolescent Psychology and Psychiatry, Erasmus MC University Medical Center Rotterdam-Sophia Children's Hospital, Rotterdam, The Netherlands. [2]The Generation R Study Group, Erasmus MC University Medical Center Rotterdam, Rotterdam, The Netherlands. [3]Department of Psychology, Harvard University, Cambridge, MA, USA. [4]Department of Radiology and Nuclear Medicine, Biomedical Imaging Group Rotterdam, Erasmus MC University Medical Center Rotterdam, Rotterdam, The Netherlands. [5]Department of Psychiatry, Massachusetts General Hospital, Harvard Medical School, Boston, MA, USA. [6]Department of Psychology, University of Minnesota, Minneapolis, MN, USA. [7]Department of Computer Science, University of Copenhagen, Copenhagen, Denmark. [8]Donders Institute for Brain, Cognition and Behaviour, Radboud University, Nijmegen, The Netherlands. [9]Radboud University Medical Center, Nijmegen, The Netherlands. [10]Leiden Institute of Advanced Computer Science, Leiden University, Leiden, The Netherlands. [11]Department of Social and Behavioral Sciences, Harvard T. Chan School of Public Health, Boston, MA, USA. [12]Department of Radiology and Nuclear Medicine, Erasmus MC University Medical Center Rotterdam, Rotterdam, The Netherlands. ✉e-mail: tiemeier@hsph.harvard.edu

nature can shed light on transdiagnostic brain-behavior associations. Distinct diagnoses may not directly map on specific underlying pathophysiology, as shared brain structures and functional connectivity have been observed across different diagnostic categories[14–16]. Multivariate methods can capture variations across brain and behaviors and identify coherent and specific brain mechanisms that cut across different diagnoses. This offers the potential for parsing possible sources of comorbidity and heterogeneity.

Given these promising features, studies of multivariate brain-behavior associations have emerged. Multivariate studies have either adopted a multiple-to-one approach (e.g., support vector machine family) using many brain features to predict cognition or diagnoses of disease, or multiple-to-multiple (doubly) approaches that can assess the covariation of many neural phenotypes (e.g., brain activity across regions) and many behavioral features simultaneously[3]. One widely used doubly multivariate method in neuroimaging is canonical correlation analysis (CCA), a technique that aims to identify the common variation across phenotypes and dissect their complex relationships into a small number of distinct components[3]. One identified component is referred as a canonical variate, which for example captures a dimension that simultaneously links multiple brain features to multiple behaviors. Several studies have implemented CCA to depict transdiagnostic brain-symptom dimensions. For instance, Xia et al.[17] applied sparse canonical correlation analysis (SCCA) to link a broad array of psychiatric symptoms to resting-state functional connectivity patterns in adolescents (age 8 to 22 and beyond, $n = 999$). Different dimensions of brain connectivity that are correlated with distinct sets of psychopathology were characterized, including mood, psychosis, fear, and externalizing behaviors[17]. This approach can identify neurally informed dimensions of psychopathology in the general population, transcending different domains of psychiatric problems and including the continuum of symptoms. This is complementary to an approach using diagnostic categorizations in a clinical sample, which has been questioned by its validity and extent of utility due to high heterogeneity within one disorder and comorbidity across diagnosed disorders[9]. Moreover, subthreshold cases which are also important for understanding psychiatric disorders[18], are not considered in the categorization approach. This is especially concerning for children, whose symptoms of psychopathology are widely recognized as dimensional[19]. The current study, therefore, adopted the dimensional approach in the general population in order to delineate neurobiological structures of child psychiatric problems.

Despite an increasing number of doubly multivariate studies being conducted, the replicability and generalizability of the techniques have come under heavy scrutiny[20,21]. One of the key elements that is largely missing from previous work, is robust external validation in a fully independent dataset (i.e., not a hold-out subsample from the same dataset). This has been widely implemented in the validation of prediction models in medical research[22,23] and recommended as a necessary step in prediction models[24]. While several non-psychiatric neuroimaging studies have established more standardized analysis pipelines[25–27], most multivariate psychiatric neuroimaging studies have not generally adopted these stringent external validation strategies[2,17,28–30].

Within the machine learning framework, an algorithm is fitted to training data and the model performance is subsequently tested on test data that is unseen and independent from the training process. This two-step procedure, in many cases embedded within a cross-validation framework, ensures the external validity (generalizability) of results[2,17,31]. In most existing studies, various forms of cross-validation have been implemented by sampling the test dataset randomly from a pool of data obtained from a single study with precisely the same imaging and assessment protocols. This means the data are often highly homogeneous in many respects, including participant sampling and data collection protocols. This step of internal validation is a reasonable start, however, understanding the real-world generalizability of a model requires a different dataset that is fundamentally distinct from the data used to train the model. This means the model must be robust to sampling and methodological differences, which is a necessity for population-level model generalizability[13]. Without this crucial step of a proper generalizability test, clinical utility will likely remain challenging. Even when care is taken to utilize a proper external dataset for testing, several problems may still exist, such as small sample sizes leading to a high potential of overfitting, and a lack of a rigorous and standardized analysis pipeline[32].

The current study aims to address these gaps by leveraging two large population-based neurodevelopmental cohorts, the Adolescent Brain Cognitive Development (ABCD) Study ($n = 4892$) and the Generation R Study[33,34] ($n = 2043$), in order to delineate robust and generalizable multivariate associations between resting-state functional magnetic resonance imaging (rs-fMRI) connectivity and child psychiatric symptoms. The ABCD study is a large, multisite study of neurodevelopment in the US. The Generation R Study (GenR) is a prospective, prenatal birth cohort in the Netherlands. As childhood and adolescence are periods of marked brain development[35] during which psychiatric problems emerge or exacerbate[36], understanding how neural mechanisms are linked to psychopathology during this time is crucial. By leveraging two large population-based samples, we were able to capture the continuum of psychiatric symptoms transdiagnostically. This enabled us to depict the brain-based dimensions of child psychopathology. Using the ABCD study as the discovery set, we applied a multivariate analysis technique, sparse canonical correlation analysis (SCCA), under a rigorous multiple hold-out framework[32,37] to identify linked brain-behavior dimensions. Importantly, the trained model in ABCD was applied and evaluated in a completely independent, external dataset (GenR) to test the out-of-study generalizability of the results. We highlight the importance of model generalizability in the context of psychiatric neuroimaging and offer several insights as to why the identification of biomarkers through these techniques remains a challenge.

## Methods
This study was not pre-registered.

### Study population
This study is embedded in two prospective cohorts of child development, the ABCD study[34] and the Generation R Study[33].

The ABCD study assesses brain development from pre-adolescence to adulthood, which was conducted across 21 study sites within the United States. Ethical approval was received from the institutional review boards of the University of California (San Diego) and each ABCD site, adhering to their Institutional Review Board approved protocols, state regulations, and local resources. Informed consent has been received from the included participants. Children at ages 9–11 were recruited as baseline and the sample is epidemiologically-informed[34]. In the ABCD cohort, resting-state functional magnetic resonance imaging (rs-fMRI) was obtained through the ABCD-BIDS Community Collection (ABCC), a community-shared ABCD neuroimaging dataset that is continually updated (https://collection3165.readthedocs.io). Both the rs-fMRI data and the behavioral assessments (data release 4.0) were retrieved from the baseline visit data of children aged 9–11 years old. Details of the study design and inclusion and exclusion criteria are detailed in previous reports[34]. Of the 9441 children whose rs-fMRI data were available, we excluded 3720 children who failed the quality control of the resting-state connectivity data (see below), 220 children with incidental findings, and 14 children with any missingness in behavioral measures and covariates. For families with multiple participants, one twin or sibling was randomly included (595 excluded). Accordingly, data from 4892 participants, of which around 7.6% had clinically relevant total problem symptom scores, were available for analysis in ABCD.

The Generation R Study is a population-based birth cohort in Rotterdam, the Netherlands. Ethical approval was obtained through the Medical Ethics Committee of Erasmus MC, University Medical Centre Rotterdam. Informed consent has been received from the included participants. Rs-fMRI data and behavioral assessments were obtained as part of the age-10 data collection which began in 2013[33]. Among the 3992 children who were scanned with MRI, 3289 rs-fMRI scans were available. We excluded children as a result of the image quality assurance protocol (see

**Table 1 | Descriptive statistics of the discovery set (example) and the external validation set**

| Discovery set | | | External validation set | |
|---|---|---|---|---|
| ABCD $n$ = 4892 | | | Generation R $n$ = 2043 | |
| | ABCD$_{Training}$ | ABCD$_{Test}$ | | |
| $N$ | 4230 | 662 | $N$ | 2043 |
| Age (years), M(SD) | 10.0 (0.6) | 10.0 (0.6) | Age (years), M(SD) | 10.1 (0.6) |
| Sex | | | Sex | |
| Female (%) | 48.9 | 48.5 | Female (%) | 52.4 |
| Race/ethnicity (%) | | | Nation of birth (%) | |
| White | 58.3 | 48.8 | Dutch | 66.1 |
| African American | 12.0 | 11.2 | Non-Dutch European | 17.3 |
| Hispanic | 19.3 | 15.1 | Non-European | 16.6 |
| Asian | 1.5 | 5.9 | | |
| Others | 8.9 | 18.0 | | |
| Parental education (%) | | | Maternal education (%) | |
| Low | 4.8 | 5.1 | Low | 2.8 |
| Medium | 38.1 | 39.1 | Medium | 34.5 |
| High | 57.1 | 55.8 | High | 62.7 |
| Child Behavior Checklist (CBCL), M(SD) | | | Child Behavior Checklist (CBCL), M(SD) | |
| Anxious/depressed | 2.5(3.1) | 2.8(3.3) | Anxious/depressed | 2.2(2.6) |
| Withdrawn/depressed | 1.0(1.6) | 1.2(1.8) | Withdrawn/depressed | 1.1(1.6) |
| Somatic | 1.5(2.0) | 1.6(1.9) | Somatic | 1.5(1.9) |
| Social | 1.5(2.1) | 1.6(2.3) | Social | 1.5(2.1) |
| Aggressive | 3.0(4.2) | 3.3(4.2) | Aggressive | 2.7(3.5) |
| Rule-breaking | 1.0(1.7) | 1.2(1.9) | Rule-breaking | 0.9(1.4) |
| Thought problems | 1.5(2.1) | 1.8(2.3) | Thought problems | 1.5(2.0) |
| Attention problems | 2.6(3.3) | 3.0(3.4) | Attention problems | 2.9(3.0) |
| Internalizing scores | 5.0(5.5) | 5.6(5.9) | Internalizing scores | 4.7(5.0) |
| Externalizing scores | 4.1(5.6) | 4.5(5.7) | Externalizing scores | 3.6(4.6) |
| Total scores | 16.9(17.2) | 19.1(18.0) | Total scores | 16.6(15.1) |

Note. Values are frequencies for categorical variables and means and standard deviations for continuous variables. The descriptive statistics for ABCD were based on one of the 30 train-test splits, other splits showed similar statistics.
*M* mean, *SD* standard deviation.

below, $n$ = 780), and children with higher than 25% missing values in the behavioral assessments ($n$ = 358). After randomly including one twin or sibling ($n$ = 108), 2043 participants (around 5.1% were clinically relevant) were included in the final sample for analysis.

### Child psychiatric symptoms

Child psychiatric symptoms were assessed using the Child Behavioral Checklist (school-age version)[38,39] in both cohorts. The CBCL is a 113-item caregiver report with eight syndrome scales (anxious/depressed, withdrawn/depressed, somatic, social, aggressive, rule-breaking, thought, and attention problems), assessing child internalizing and externalizing problems. Internalizing problems reflect a variety of inner-directed symptoms, such as anxiety, withdrawal, or depression, while externalizing problems incorporate outer-directed symptoms, such as aggression and rule-breaking behaviors[40]. The CBCL was administered in both cohorts and the primary caregivers answered 113 items on a three-point Likert scale (not true, somewhat true, very true) for problems in the past six months. The current analyses relied on raw scores from the CBCL, as is recommended by the instrument authors to preserve the full range of variation[39]. Items belonging to a given syndrome scale were summed. In the case of missing items, if the missingness was less than 25%, a sum score was created accounting for missing items. Higher scores represent more problems. The raw sum scores of each syndrome scale were within the normal range both in ABCD and GenR. For the detailed statistics for the eight syndrome scales in ABCD and Generation R, see Supplementary Table 9 and Table 1.

### fMRI image acquisition and preprocessing

Rs-fMRI data in the ABCD datasets were acquired from 3-Tesla scanners from three manufacturers (Siemens Prisma, Philips, and General Electric (GE) 750) across 21 study sites. For detailed MRI acquisition parameters, please see Supplementary Table 1. In GenR, rs-fMRI imaging data were collected on a single-site 3 Tesla GE Discovery MR750w MRI System scanner. Details of the MRI acquisition parameters are presented in Supplementary Table 2.

The BIDS data were preprocessed with the fMRIPrep pipeline[41] both in ABCC (version 20.2.0) and GenR (version 20.2.7). Briefly, structural MRI data first underwent intensity normalization to account for B$_1$-inhomogeneity and brain extraction, followed by nonlinear registration to MNI space and FreeSurfer processing. Functional MRI data then underwent volume realignment with MCFLIRT (FSL). BOLD runs were then slice-time corrected with 3dTshift (AFNI), followed by co-registration to the corresponding T1w reference. Data were ultimately resampled to FreeSurfer fsaverage5 surface space. Of note, the first 5-minute run of resting-state data in ABCD was extracted to further optimize comparability with GenR (5 min 52 s).

### Parcellation and whole-brain connectivity estimation

The connectivity estimation procedure was identical in ABCD and GenR and was performed using Python (version 3.9.0). Whole-brain functional connectivity matrices were calculated and mapped onto the Gordon cortical parcels[42] and FreeSurfer subcortical segmentation[43], yielding 349 distinct parcels consisting of 333 cortical and 16 subcortical regions. Briefly, after removing the first 4 volumes from each dataset to ensure magnetic stability, the BOLD signals were averaged across all voxels in each cortical and subcortical region. Then the extracted time series were adjusted for CSF and white matter signals (plus their temporal derivatives and quadratic terms), low-frequency temporal regressors for high-pass temporal filtering, and 24 motion regressors (6 base motion parameters + 6 temporal derivatives + 12 quadratic terms). Pearson correlation was applied to estimate the temporal dependence between the residualized regional time series and the estimated connectivity was Fisher z-transformed, resulting in a symmetric 349 × 349 functional connectivity matrix for each participant.

### Quality controls of the scans

In the ABCC datasets, only data that passed the initial acquisition Data Analysis Imaging Center (DAIC) quality control were included. Briefly, at the time of scanning, quality control was performed by scan operators with a binary pass or fail. In our study, participants were further excluded based on the ABCD recommended guidelines (imgincl_rsfmri_include = 1), which involve raw and postprocessing quality control, passed FreeSurfer QC, had more than 375 rs-fMRI frames after censoring, and other cut-off scores (see ABCD Recommended Imaging Inclusion), 1310 participants were excluded due poor quality. We additionally excluded 2410 participants with excessive motion (mean framewise displacement (FD) higher than 0.25 mm)[44], and 220 participants with clinically relevant incidental findings.

In Generation R, the following exclusion criteria were applied to screen eligible participants: (1) scans with major artifacts (e.g., dental retainers, or other scan-related artifacts), (2) scans lacking whole-brain coverage (e.g.,

missing large portions of the cerebrum or cerebellum from the field of view), and (3) scans with excessive motion (mean framewise displacement (FD) higher than 0.25 mm or having more than 20% of the volumes with an FD higher than 0.2 mm)[44]. Moreover, the accuracy of co-registration was visually inspected by merging all co-registered images into a single 4D Nifti image and scrolling through the images. 583 scans with poor quality were excluded in total.

## Covariates

In ABCD, child age, sex, race/ethnicity, parental education, and data collection site were used as covariates. Demographic information (child age, sex, race/ethnicity, and parental education) were assessed by caregiver-report questionnaires. The original 21-category parental education was recoded into three categories to make it comparable with Generation R: 1st to 12th grade, high school/GED/college, and Bachelor's degree or higher.

In Generation R, similar covariates were included except for study sites, including age of children when undergoing the MRI scanning, sex, child national origin, and maternal education. All the information was obtained from questionnaires completed by caregivers. Child national origin was defined based on the birth country of the parents and was coded into three categories: Dutch, non-Dutch European, and non-European. Maternal education, an indicator of socioeconomic status, was recoded into three categories: maximum of three years secondary school, more than three years general secondary school; intermediate vocational training, and Bachelor's degree or higher[45]. Missing values were imputed by using Expectation-Maximization imputation as the proportion of missing values was smaller than 1% of the current Generation R data set[46].

## Child cognitive ability

Child cognitive ability data was retrieved from NIH Toolbox age-corrected standard scores of fluid intelligence (adaptive problem-solving), crystallized intelligence (knowledge acquisition from experience), total cognition scores (overall cognition composite scores), and matrix reasoning scaled scores (non-verbal reasoning) from the Wechsler Intelligence Scale for Children-V (data release 4.0)[47,48].

## Analysis framework

The current study implemented a multiple hold-outs framework that aims to increase the generalizability of the analysis and inspect potential sampling bias of training and hold-out datasets[32]. We used ABCD as the discovery set ($n = 4892$), in which all analyses were conducted (trained) and tested. The ABCD discovery set was randomly split into a training set consisting of 18 sites ($ABCD_{Training}$) and a test set consisting of 3 sites ($ABCD_{Test}$). In this way, subjects in the $ABCD_{Training}$ and $ABCD_{Test}$ sets were ensured to be entirely from different sites, approaching the true out-of-sample setting (Fig. 1). To reduce sampling biases, the split procedure was repeated 30 times, resulting in 30 pairs of independent train-test sets. This is suggested as a multiple hold-out framework[32,37], which can examine and reduce possible fluctuation of results in different training and hold-out sample split. Importantly, all the analyses in $ABCD_{Training}$ sets and $ABCD_{Test}$ sets were fully separated to protect the results from data leakage (Fig. 1), including the residualization of brain data, weighted PCA, and SCCA model. Specifically, the residualization was separately done in $ABCD_{Training}$ sets and $ABCD_{Test}$ sets, and the weighted PCA was first implemented in $ABCD_{Training}$ sets, then the PCA eigenvectors retrieved from $ABCD_{Training}$ set were applied to $ABCD_{Test}$ set to derive brain PCs. Next, the SCCA model was trained in $ABCD_{Training}$ set, where the penalty parameters of the SCCA models were selected in 100 further random splits of training (80% of $ABCD_{Training}$ set) and validation sets (20% of $ABCD_{Training}$ set). After fitting the model with the optimal hyperparameters in the $ABCD_{Training}$ set, the out-of-sample model generalizability was evaluated by projecting the CBCL and brain PCs loadings trained in $ABCD_{Training}$ set to $ABCD_{Test}$ set. In the final step, Generation R, which has ascertained a large early adolescent sample with very similar measures, was used as an independent external validation set ($n = 2043$). We characterized two approaches of external validation (see

Out-of-study generalizability test in Generation R), allowing us to estimate the out-of-study generalizability of the findings from ABCD. Moreover, we did several explorations of the identified brain canonical variates in ABCD. First, we tested whether the identified brain canonical variates were associated with child cognitive ability at the age of 10. Second, we investigated whether we could find distinct subgroups/clusters of children based on the identified brain canonical variates.

## Dimensionality reduction

Prior to SCCA analysis, the upper triangle of the 349 × 349 functional connectivity matrix was flattened, resulting in 60,726 connectivity features for each participant. Connectivity values were residualized to ensure the above-mentioned covariates did not influence the results[17]. As the high-dimensional nature of the connectivity features could lead to considerable overfitting in SCCA, weighted principal component analysis (PCA) was applied to reduce the connectivity features into principal components (PCs) that aggregated the information of the data[49]. This PCA-CCA framework has been used extensively and has shown good performance[16,50].

While traditional PCA only considers the structure of the brain data, the weighted PCA uses the relationship between the brain and behavioral data in dimensionality reduction to identify a relatively small number of PCs carrying information from the phenotypes of interest[49]. This ensures the variability in the functional connectivity data most related to behavioral and emotional problems will be captured in the PCs. To achieve this, we rescaled the connectivity data according to a rank-based weighting scheme, which depends on the sum of CBCL scores. The weight assigned to each subject was determined by the rank of their total CBCL score. The rank-based pre-weights were calculated as follows:

$$\tilde{w}_i = \ln n - \ln r_i$$

Where n is the number of data points and r is the ranking. We normalized the pre-weights by $w_i = \tilde{w}_i / \sum \tilde{w}_i$, and the original connectivity data was demeaned and adjusted with the corresponding normalized weights. We then submitted the adjusted connectivity matrix to PCA, and the eigenvectors (variable loadings) of PCA were extracted and multiplied with the original connectivity matrix, resulting in a new, dimensionally reduced weighted connectivity matrix. To further protect against overfitting in subsequent analyses, a selection of PCs was made, namely the first 100 principal components[16].

## Sparse canonical correlation analysis

To delineate multivariate relationships between functional connectivity and child psychiatric problems, we applied sparse CCA (SCCA), an unsupervised learning technique that can simultaneously evaluate the relationships between two sets of variables from different modalities[3]. SCCA imposes both $l_1$-norm and $l_2$-norm penalty terms, an elastic net regularization combining the LASSO and ridge penalties, to high-dimensional datasets and achieves sparsity of the solution[51]. This method is more stable, more robust to deviation from normality, and does not have the main constraint of classic CCA: the number of observations should be larger than the number of variables[17]. Specifically, given two matrices, $X_{n \times p}$ and $Y_{n \times q}$, where n is the number of participants, p and q are the number of variables (e.g., CBCL scores and brain PCs, respectively), SCCA aims to find u and v (canonical loading matrices) that maximize the covariance between Xu and Yv. Xu and Yv are canonical variates that are the low dimensional representation of brain and behavioral measures.

## Selection of penalty parameters

Using the extracted 100 components after dimensionality reduction, we first determined the optimal penalty parameters before fitting the SCCA. In order to identify the best set of penalty parameters for the SCCA of functional connectivity and behavioral features, we used a repeated resampling procedure of the $ABCD_{Training}$ set[32,37] (Fig. 1). Specifically, we first split the $ABCD_{Training}$ set further into penalty parameter training (80%) and

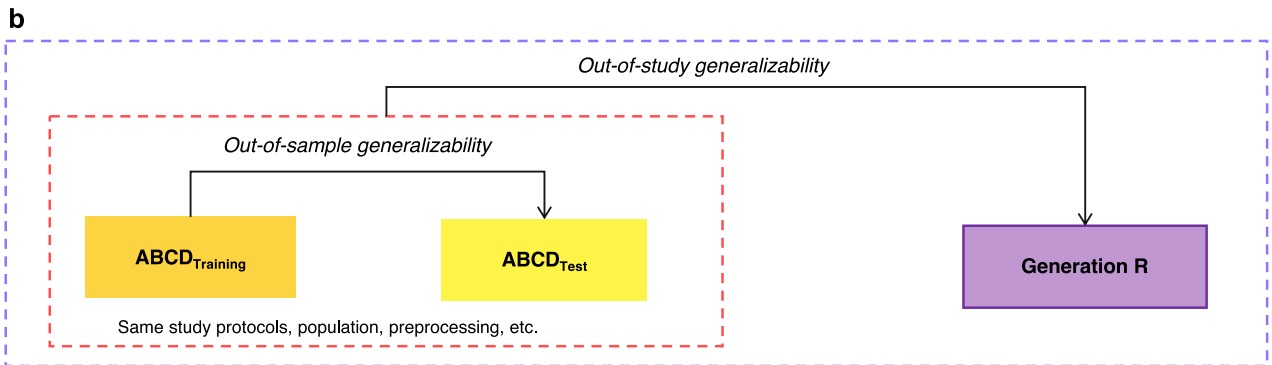

**Fig. 1 | Multivariate brain-behavior associations analysis pipeline. a** ABCD was the discovery set and Generation R as the external validation set. The discovery set was divided into training and test sets 30 times, resulting in 30 train-test pairs in ABCD. The eigenvectors of PCA from the ABCD$_{Training}$ set were applied to ABCD$_{Test}$ set to calculate the principal components, then the canonical loadings obtained from the ABCD$_{Training}$ set were projected to ABCD$_{Test}$ set to compute the out-of-sample generalizability. Similarly, weight vectors of SCCA from the ABCD$_{Training}$ set were then directly applied to Generation R to assess the out-of-study generalizability of the model. We also implemented the qualitative replication approach, in which we train the CCA model independently in Generation R and compare the results across the two cohorts. Note that the sample size in ABCD is an example from one train-test split. **b** Out-of-sample generalizability within ABCD and out-of-study generalizability in Generation R.

validation set (20%) 100 times, resulting in 100 pairs of training and validation sets. Next, a grid search between 0 and 1 with increments of 0.1 was used to determine the combination of penalty parameters ($l_1$ and $l_2$) that show the best performance[17]. For each combination of penalty parameters, we fitted the SCCA model in the training set, projected the canonical loadings extracted from the training set (u and v) on the validation set, and then calculated the canonical correlations. The optimal combination of penalty parameters was chosen based on the highest first canonical correlation of the validation set averaged across 100 splits. To improve the interpretability of the behavioral loadings, the penalty parameter for behavioral measures (eight CBCL syndrome scores) was constrained to be larger than 0.5.

## Fitting SCCA model and significance test

After the selection of optimal penalty parameters, the SCCA model was fitted to $ABCD_{Training}$ set with the chosen parameters. The resulting weight vectors (canonical loadings) from $ABCD_{Training}$ set were then projected onto brain PCs and CBCL scores of $ABCD_{Test}$ set (after first deriving brain PCs in the $ABCD_{Test}$ set by applying the eigenvectors of the weighted PCA from $ABCD_{Training}$ set). This process yielded the canonical correlations in the $ABCD_{Test}$ set, reflecting the within-cohort out-of-sample generalizability of the SCCA model. To determine the statistical significance of each canonical correlation, a permutation testing procedure was applied both in the $ABCD_{Training}$ and $ABCD_{Test}$ sets. In the permutation test, the rows of the behavioral data were shuffled to disrupt the relationship between the brain connectivity features and the behavioral features, while the brain connectivity matrix held constant[20]. We performed 2000 permutations, building a null distribution of each canonical correlation. The $p$-value (two-sided) of the permutation test is defined as the number of null correlations that exceeded the correlations estimated on the original, un-shuffled dataset. The same set of penalty parameters was used in each permutation. False Discovery Rate (FDR) was used for multiple testing correction. Only canonical variates surviving permutation testing ($p < 0.05$) were selected for further analysis.

## Stability of SCCA model

The classical CCA has been found to be unstable at times and fails to converge when the samples-to-feature ratio is small[52]. To investigate the sampling variability of the canonical loadings and inspect the features that consistently contributed to each canonical variate in the SCCA model, 1000 bootstrapping subsamples (sample with replacement) were generated. The distribution of canonical loadings in this procedure allows us to inspect the stability and sampling variability of the SCCA model. This was done in one randomly selected train-test split. As arbitrary axis rotation could be induced by bootstrapping, leading to the changes in the order of canonical variates and the signs of the canonical weights, we matched the order of canonical variates based on the CBCL loadings we derived we derived from the original datasets[17].

## Associations with cognitive ability

To further validate the canonical variates we found, we tested whether the identified brain canonical variate scores were associated with child cognitive ability at the age of 9 to 10 in the ABCD cohort. The brain canonical variate scores were computed by multiplying the raw brain PCs with the derived brain canonical loadings. We separately modeled the relationship between each canonical variate score of brain connectivity and the cognitive ability of the participants with linear regression models adjusted for all covariates.

## Out-of-study generalizability in Generation R

CCA is vulnerable to overfitting and the external validity of the canonical variates should be carefully investigated[32,52]. In the current study, we tested the external validity of the findings from the ABCD discovery set in an independent dataset: Generation R. We utilized two approaches to test the external validity: the qualitative replication and the gold-standard generalizability test. Qualitative replication means repeating the analyses in

different settings and observing the similarities of findings across studies, while generalization means the same statistical model successfully makes predictions in different populations. Replication provides evidence of important correlations, and further generalizability tests are conducted to provide a realistic possibility for extending these discoveries into clinical applications. In the qualitative replication, the SCCA model was independently trained on Generation R, yielding another set of canonical loadings. To evaluate generalizability, we multiplied the input data of GenR with the canonical loadings derived from ABCD. The derived canonical variate scores were then correlated with the canonical variate scores trained in GenR (input data of GenR multiplied with loadings trained in GenR) using Pearson's correlation. This was also done in a reverse direction, where we multiplied the input data of ABCD with canonical loadings derived from GenR and calculated their correlations with the canonical variate scores trained in ABCD[53]. The significance of correlations was determined by permutation tests ($n = 5000$) and significance in both directions was considered as generalizable[53]. Similar to what is described above for the out-of-sample generalizability test in ABCD, in another, more standard practice in machine learning studies, the gold-standard generalizability test, we projected the SCCA canonical loadings of $ABCD_{Training}$ directly on brain PCs and CBCL scores of Generation R. The canonical correlations were ultimately calculated and assessed with permutation testing.

## Sensitivity analysis

We conducted several sensitivity analyses to ensure our results were not biased by the analytical approach we selected. First, we implemented 5-fold cross-validation (18 sites for training and 3 sites for testing) without any repeated study sites in the five $ABCD_{Test}$ sets. This aimed to rule out the possibility that repeated study sites in the original 30 $ABCD_{Test}$ sets might inflate the results in ABCD. Second, we performed a standard PCA without the weighting scheme before SCCA to test whether the overfitting was the result of the weighted PCA. Third, we applied traditional CCA after a standard PCA to see how the sparsity of the model influenced the results. Fourth, we included different numbers of brain principal components to inspect possible fluctuation of results due to the dimensionality of brain data, which might be one of the reasons for overfitting. Fifth, we used the non-harmonized brain data to investigate the potential impact of the preprocessing procedure on the generalizability of results.

## Reporting summary

Further information on research design is available in the Nature Portfolio Reporting Summary linked to this article.

## Results

A total of 6935 (see "Methods" for inclusion criteria) resting-state functional MRI scans from the multisite ABCD Study (ages 9-to-11 years from 21 study sites) and the single-site GenR Study (ages 9-to-12 years) were summarized using the 349 region Gordon parcellation[42]. After several salient functional MRI confounders were regressed out (e.g., motion, see "Methods"), functional time courses from the different regions (333 cortical, 16 subcortical) were used to construct connectivity matrices for each individual by correlating the time courses pair-wise across all regions. To mitigate possible overfitting problems, the connectivity matrices underwent dimensionality reduction by principal component analysis (PCA) with a weighting scheme (see "Methods"). Psychiatric symptoms of children were assessed using the Child Behavioral Checklist (CBCL, school-age version)[34], a caregiver report with eight syndrome scales (anxious/depressed, withdrawn/depressed, somatic, social, aggressive, rule-breaking, thought, and attention problems). To improve the generalizability of the results, the ABCD sample was randomly split into a training set consisting of 18 sites ($ABCD_{Training}$) and a test set consisting of 3 sites ($ABCD_{Test}$). The split procedure was repeated 30 times to reduce sampling bias, resulting in 30 pairs of independent train-test sets. Importantly, the analyses in $ABCD_{Training}$ and $ABCD_{Test}$ sets were fully separated to help minimize the potential for data leakage (Fig. 1). Generally, $ABCD_{Training}$ and $ABCD_{Test}$

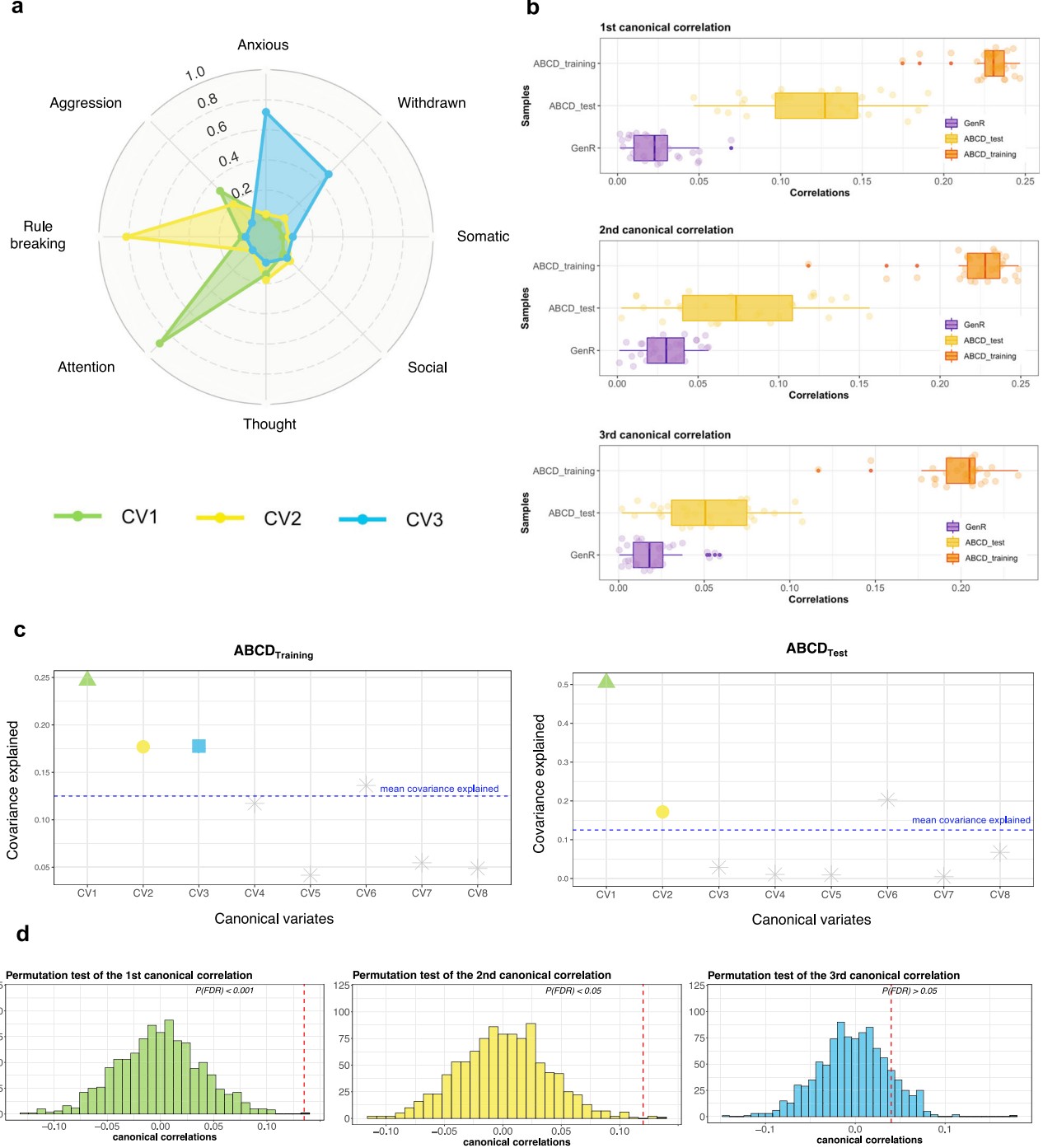

**Fig. 2 | Two associated dimensions of brain connectivity and CBCL scores in ABCD. a** The canonical loadings of CBCL syndrome scores of the first three canonical correlations in the ABCD$_{Training}$ sets. The loadings were averaged across 30 train-test splits. Green represents CV1, yellow represents CV2, and blue represents CV3. **b** The median (center line), 25th percentile and 75th percentile (box limits) of the first three canonical correlations across 30 train-test splits in ABCD and Generation R. The orange boxplots are for ABCD$_{Training}$ sets, the yellow boxplots are for ABCD$_{Test}$ sets, and the purple boxplots are for Generation R. **c** Covariance explained in the training and test sets (example from one train-test split). The number of canonical variates in the ABCD$_{Training}$ set that was put into the permutation test was selected based on the ones that were larger than the mean covariance explained. **d** Permutation tests in the ABCD$_{Test}$ sets (example from one split). The first canonical variate was largely generalizable in ABCD$_{Test}$ set across the 30 train-test splits, the second to a less extent, and the third was not generalizable. The red dotted lines represent the canonical correlations in the unshuffled data. CV1: canonical variate 1, CV2: canonical variate 2, CV3: canonical variate 3.

sets were matched on age, sex, race/ethnicity/parental education, and psychiatric symptoms (Table 1).

### Initial derivation of brain-behavior dimensions in ABCD

Using ABCD$_{Training}$ set (18 of the 21 ABCD study sites) to train the model (repeated 30 times), three brain-symptom dimensions (canonical variates) were identified from the functional connectivity data (100 principal components, with averaged explained variance of 61.9% across 30 train-test splits) and the psychiatric symptom data (raw sum scores of the 8 CBCL syndrome scales) using an elastic net combining LASSO and ridge penalties within SCCA ($r_1 = 0.23, r_2 = 0.22, r_3 = 0.20, ps < 0.001$; correlations averaged across 30 splits, see Fig. 2a, Supplementary Table 3).

## Out-of-sample generalizability of brain-behavior dimensions in ABCD

Next, to ascertain the out-of-sample generalizability of the model, the remaining 3 ABCD study sites (ABCD$_{Test}$ set, repeated 30 times) were used. By applying the eigenvectors of the weighted PCA from the ABCD$_{Training}$ set along with the resulting weight vectors (canonical loadings) from the SCCA of the ABCD$_{Training}$ set, the model parameters were effectively projected onto the functional connectivity data and psychiatric symptom data from the ABCD$_{Test}$ set. This 'gold standard' process for evaluating out-of-sample model performance yielded canonical correlations in the ABCD$_{Test}$ set. Overall, we found evidence that the first canonical correlation was robustly identified across train-test splits, the second to a lesser extent, and the third failed to be validated, in the ABCD$_{Test}$ sets. Specifically, the first dimension was validated across 24 out of 30 splits ($r_1 = 0.12$, $ps < 0.05$, Supplementary Table 3). This brain-symptom dimension captured the correlates between attention problems and connectivity patterns in connectivity networks involved in higher-order functions (salience, cingulo-opercular, and frontoparietal network)[54], visual-spatial attention network (parietal occipital, medial parietal)[55], motor networks, and default mode network (Fig. 3a, c). The second dimension was only evident in 12 out of 30 splits ($r_2 = 0.07$, $ps < 0.05$, Supplementary Table 3). This linked dimension delineated a relationship between aggressive and rule-breaking behaviors and connectivity patterns in similar networks involved in higher-order and visual-spatial attention functions, with a larger contribution from subcortical areas and motor networks (Fig. 3b, d). Interestingly, across two linked dimensions, the salience, parietal occipital, motor, and cingulo-opercular networks were overlapped. The third dimension was only observed in 5 of the 30 splits ($r_3 = 0.05$, $ps > 0.05$; all three presented correlations were averaged across 30 splits, and $p$-values were corrected for multiple testing using the false discovery rate), and thus was not considered a stable, internally valid dimension. Interestingly, when splitting the ABCD sample into train/test sets differently (i.e., allowing all study sites to be represented in both training and testing sets), the first two canonical variates were more stable and demonstrated a smaller decrease in the magnitudes of canonical correlations from training to test sets (Supplementary Table 4). These results suggest the SCCA will likely be more prone to overfitting when training and testing sets contain data from all ABCD study sites.

## Stability of the brain-based dimensions of child psychiatric symptoms in ABCD

To further interpret the characteristics of each canonical variate and the stability of canonical loadings, 1000 bootstrap subsamples were generated to identify the CBCL syndrome scores and brain PCs that are consistently heavily loaded for different canonical variates (see "Methods"). The variability of the first three canonical correlations, CBCL canonical loadings and brain connectivity canonical loadings are presented in Fig. 4. Importantly, the three canonical correlations decreased considerably in the ABCD$_{Test}$ set compared to the ABCD$_{Training}$ set (Fig. 4c). While relatively stable contribution from the CBCL syndrome scores was observed, the instability of rs-fMRI canonical loadings manifested through more variability and less clear patterns in the canonical loadings for brain PCs (Fig. 4a, b). Thus, despite being robust, the dimensions were to some extent overfit in the ABCD$_{Training}$ sets.

## Out-of-study generalizability in a fully independent sample

Although the ABCD Study is a multisite study, it is a highly harmonized dataset in the context of the imaging and behavioral data, and also likely has sampling characteristics that are specific and uniform across sites. Thus, to test the out-of-study generalizability of the results we obtained in ABCD, we use the Generation R Study (GenR) as an independent external validation set. The GenR is a single-site population-based birth cohort in Rotterdam, the Netherlands[33], which has ascertained a large, early adolescent sample with very similar measures as the ABCD Study. The resting-state connectivity data from ABCD and GenR

were highly harmonized by undergoing the same preprocessing pipeline and methods of calculating connectivity matrices (see "Methods"). We included 2043 children at the age of 10 with good-quality resting-state connectivity data and less than 25% missingness in the CBCL assessment. We characterized two approaches of external validation. One approach is the more commonly used qualitative replication[28,29], where a new SCCA model was independently trained on Generation R and the similarities of results between cohorts were compared. Another is the gold-standard generalizability test, where we projected the SCCA model weights of the ABCD$_{Training}$ set onto the first 100 brain PCs (explained variance 61.8%) and CBCL syndrome scores of Generation R (see "Methods").

In the gold-standard generalizability test, we only observed the first canonical correlations survived permutation testing in 1 of the 30 train-test splits (Supplementary Table 3). All other canonical correlations did not survive in GenR when we used the SCCA models that were trained in ABCD ($r_1 = 0.03$, $r_2 = 0.03$, $r_3 = 0.02$, $ps > 0.05$; correlations averaged across 30 train-test splits, Fig. 2b, Table 2). These results are highly consistent when we performed a standard PCA without the weighting scheme before SCCA (Supplementary Table 5), when we applied traditional CCA without sparsity (Supplementary Table 6), when we put different numbers of brain PCs (i.e., 50 and 200) in the model (Supplementary Tables 7 and 8, Supplementary Fig. 1), and when we implemented the model using non-harmonized brain data (Supplementary Table 10).

In the other, more commonly used qualitative replication in doubly multivariate studies[17,28,53], a new SCCA model was trained in Generation R, yielding another set of canonical loadings. The correlations between the two sets of canonical loadings (those from ABCD and GenR) were considered as a proxy for out-of-study generalizability (see details in "Methods"). To make the results more comparable, we trained the SCCA model in Generation R using the same sparsity parameters that were most selected in ABCD. After the permutation test, four significant canonical variates were identified in Generation R. Specifically, three similar canonical correlations were also observed in Generation R (Fig. 5a, b), showing cross-cohort correlations of $r = 0.96$–$0.97$ ($p < 0.001$) for the CBCL canonical variate scores of attention problems, $r = 0.92$–$0.94$ ($p < 0.001$) for aggressive and rule-breaking behaviors, and $r = 0.83$–$0.84$ ($p < 0.001$) for anxious and withdrawn behaviors. Several of the most important brain connectivity networks involved were overlapped with ABCD (Supplementary Fig. 2). For instance, both in ABCD and GenR, the salience, parietal occipital, and motor networks were the most crucial contributors to the association of attention problems and brain connectivity. Similarly, the motor, auditory networks, and subcortical areas contributed most to the correlates of aggressive and rule-breaking behaviors and brain connectivity. However, the contributions of other networks, or the collective effects of all the networks, were not the same across the two cohorts (Supplementary Fig. 2c, d), suggesting that variability in the brain phenotypes is underlying the poor out-of-study generalizability. The remaining canonical correlation found in Generation R was related to social and anxious problems and did not overlap considerably with those observed in ABCD.

## Further exploration of brain canonical variates

Being a popular dimensionality reduction or multimodal fusion method in neuroimaging studies, the brain canonical variate scores from SCCA are often used further as the input for other statistic models or clustering algorithms[3,56]. In the final step, we first explored whether the two identified brain canonical variates are associated with the cognitive ability of children at 10 years old in ABCD. Cognitive ability data was retrieved from NIH Toolbox age-corrected standard scores of fluid intelligence, crystallized intelligence, total cognition scores, and matrix reasoning scaled scores (non-verbal reasoning) from the Wechsler Intelligence Scale for Children-V (data release 4.0)[47,48]. We separately modeled the relationship between each canonical variate of brain connectivity and the cognitive ability of the participants with linear regression models adjusted for all covariates. The results showed that the first brain connectivity canonical

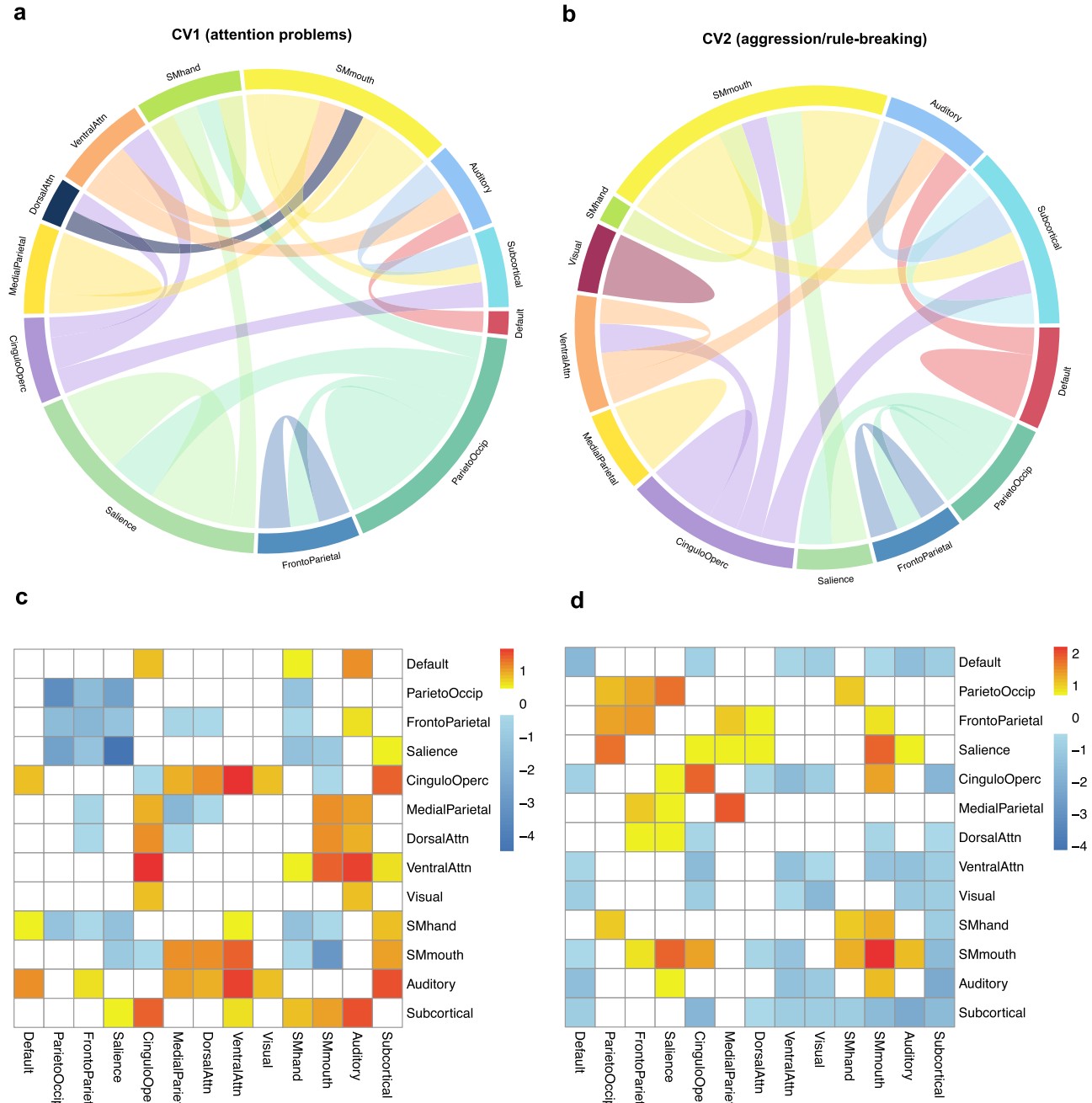

**Fig. 3 | Resting-state connectivity modules involved in the two identified associations in ABCD. a, b** The top 20% of the connectivity patterns that contributed most for each of canonical variate. The outer labels represent the names of network modules. The thickness of the chords showed the importance of different network modules. The contribution of each connectivity feature was determined by computing the correlations between the raw connectivity features and the canonical variate scores of the brain connectivity extracted from the SCCA model (calculated by canonical loadings averaged across 30 train-test splits multiplied with the brain PCs of the whole sample of ABCD), indicating the importance of each connectivity feature. After calculating the contribution of each connectivity feature, we summarized the contributions based on pre-assigned network modules and calculated the within and between-network loadings. **c, d** The connectivity patterns associated with the first two canonical variates. This was based on the z-scores of the within- and between-network loadings we calculated.

variates (attention problems) were associated with fluid and crystallized intelligence, and total cognition scores, while the second brain canonical variate score (aggressive/rule-breaking behaviors) was only associated with crystallized intelligence and total cognition scores (Table 3). We also examined the possibility of meaningful clustering solely based on the brain canonical variates we found, frequently referred to as 'biotypes' (see Supplementary Methods). We did not find distinct clusters with disparate clinical profiles along the two brain canonical dimensions (Supplementary Fig. 3, Supplementary Table 11).

## Discussion

Several studies have highlighted the intriguing potential of multivariate brain-behavior associations, but the lack of replicability of results has hampered the identification of robust neurobiological mechanisms underlying psychiatric problems[4,20]. To improve the robustness and generalizability of brain-behavior associations in a fully independent sample, which is largely sub-optimally done or absent in previous research in multivariate psychiatric neuroimaging literature, the present study implemented the SCCA method and tested out-of-sample generalizability in one cohort

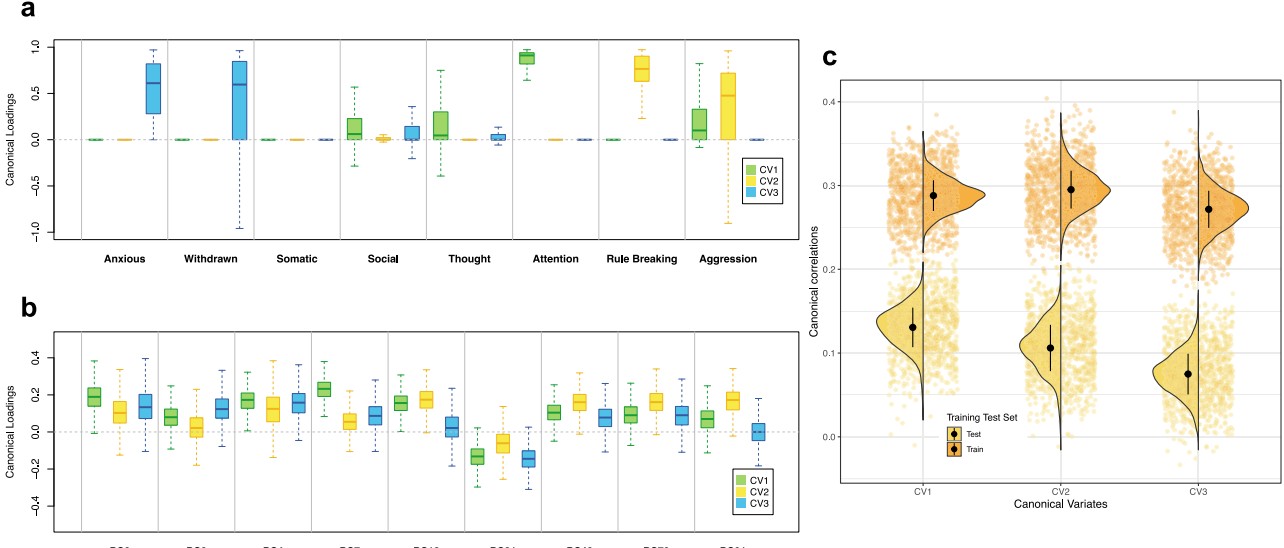

**Fig. 4 | Stability of canonical correlations and canonical loadings in ABCD (example). a** The variability for the canonical loadings of CBCL syndrome scores across 1000 bootstrap subsamples. The center line is median, the upper quantile is 75% and the lower quantile is 25%. **b** The variability for the canonical loadings of brain PCs across 1000 bootstrap subsamples. The PCs presented here were selected based on the intersection of top 10 most important PCs for the first three canonical variates. **c** The variability of the first three canonical correlations in ABCD$_{Training}$ and ABCD$_{Test}$ set. The black dot is mean, and the vertical black line is standard deviation. Note that the bootstrap subsampling is conducted in one of the 30 train-test splits. CV1: canonical variate 1, CV2: canonical variate 2, CV3: canonical variate 3.

evaluated out-of-study generalizability in a fully external cohort. Robust multivariate brain-psychiatric symptom associations in children were observed, however, a gold-standard test of the generalizability of the findings in another cohort was largely negative. While these results reinforce previous work demonstrating the potential for brain-based dimensions of psychiatric problems, they also highlight the problem of the generalizability of findings in psychiatric neuroimaging studies, especially in the general population.

In ABCD, we identified two brain-symptom dimensions that were consistently validated in the out-of-sample test sets, indicating robust within-study (internally valid) multivariate brain-symptoms associations. The first brain-symptom dimension mapped on attention problems, reflecting hyperactive and inattentive behaviors[57]. Several connectivity networks loading highly on this dimension, such as salience, parietal occipital, and medial parietal networks, have been shown to be involved in attention deficit hyperactivity disorder (ADHD) across studies[58,59]. These networks have been implicated in deficits of top-down executive control, attention, and spatial working memory in children with ADHD[55,58–60]. The second brain-based dimension was centered on aggressive and rule-breaking behaviors. Several similar connectivity networks observed with the first dimension were observed with the second. The motor and visual network, which were involved in the hyperactivation of the motor system and the tracking of external stimuli[61,62], played a more important role here. Overall, the first two brain-symptom dimensions reflect differences in brain connectivity that are related to child externalizing problems.

**Table 2 | Failed gold-standard generalizability test in Generation R**

| Canonical correlations | ABCD | | Generation R |
|---|---|---|---|
| | training set | test set | |
| $r_1$ | 0.23 | 0.12* | 0.03 |
| $r_2$ | 0.22 | 0.07* | 0.03 |
| $r_3$ | 0.20 | 0.05 | 0.02 |

Note. Canonical correlations in ABCD were averaged across the 30 train-test splits.
r1: *p < 0.05 in 24 train-test splits. r2: *p < 0.05 in 12 train-test splits.
r1 Generation R: p < 0.05 in 1 train-test split.

The two identified brain-based dimensions were further validated by their associations with child cognitive ability, which is in line with results in behavioral studies showing associations between externalizing problems and measured intelligence[63]. Throughout the two dimensions, the motor, parietal occipital, cingulo opercular, and salience networks were consistently involved, indicating shared patterns of functional connectivity across different symptom-defined profiles. The specific patterns alongside the common (shared) patterns of functional connectivity across three dimensions might implicate a similar general-psychopathology-factor structure of child psychopathology identified by both clinical and genetic measures[64,65] or could reflect non-specificity in brain-behavior associations. Indeed, another study also observed an association between the default mode network and the general psychopathology(p) factor in children[66]. This work, therefore, adds a piece to solving the puzzle of high comorbidity and heterogeneity of child psychiatric problems.

While we discovered two brain-symptom dimensions in ABCD, the out-of-study generalizability in Generation R presented a complex challenge. Psychiatric neuroimaging studies employ varying approaches to test generalizability, and thus demonstrate varying degrees of external validity. One approach consists of repeating the analysis pipeline and training a new model in data that were previously 'unseen' by the doubly multivariate algorithm, and then correlating the model weights across studies. This is often referred as 'replication', in a test set from the same large participant pool or an external dataset. Similarities of behavioral or brain loadings are usually used to indicate a successful replication[17,28]. In the present study, we observed highly similar behavioral dimensions when training the SCCA model independently in Generation R. The robust dimensions observed in the discovery set (ABCD), alongside the similar behavioral dimensions observed in the qualitative replication, lend support for the internal validity of the brain-behavior dimensions. Therefore, the results are convincing in the general context of underlying dimensional neurobiology. However, even though this route of replication is a valuable way to demonstrate whether the brain-behavior associations exist from an empirical perspective, precisely how one can define a successful replication based on the qualitative or quantitative similarities between results remains a non-trivial challenge for the field.

Importantly, the more robust, gold-standard generalizability test, where the weight coefficients of the SCCA model from the discovery set (ABCD) are projected to the independent sample (Generation R), was not

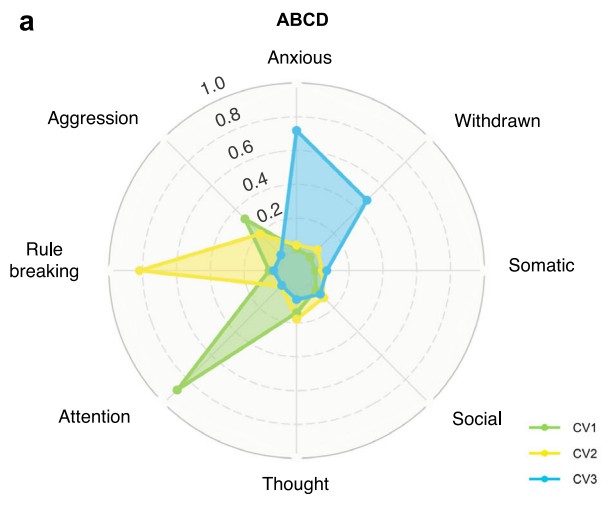

**Fig. 5 | CBCL canonical loadings in ABCD and Generation R in qualitative replication. a** The canonical loadings of CBCL syndrome scores in ABCD, averaged across 30 train-test splits. **b** The canonical loadings of CBCL syndrome scores in Generation R. CV1: canonical variate 1, CV2: canonical variate 2, CV3: canonical variate 3, CV4: canonical variate 4.

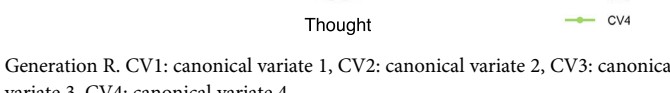

**Table 3 | Associations between the two brain canonical variate scores and cognitive ability (*n* = 3968)**

| | Fluid intelligence | | Crystallized intelligence | | Matrix reasoning | | Total cognition | |
|---|---|---|---|---|---|---|---|---|
| | B (95% CI) | P | B (95% CI) | P | B (95% CI) | P | B (95% CI) | P |
| CV1 | −0.04 [−0.07, −0.01] | 0.01 | −0.04 [−0.07, −0.01] | 0.003 | −0.001 [−0.03, 0.03] | 0.93 | −0.05 [−0.08, −0.02] | <0.001 |
| CV2 | −0.03 [−0.06, 0.00] | 0.05 | −0.04 [−0.07, 0.02] | 0.003 | 0.003 [−0.03, 0.03] | 0.82 | −0.05 [−0.08, −0.02] | 0.002 |

Separate linear regression analysis of cognitive ability and the two brain canonical variate scores. Betas are standardized. All the models were adjusted for child age, child sex, race/ethnicity, parental education, and scanning sites. After excluding the participants with missing values in any of the four cognitive abilities, the final sample size of this analysis is 3968. CV1: brain canonical variate 1, CV2: brain canonical variate 2.

successful. It is clear that the common, yet suboptimal, practices of replication mentioned above provide an important role in uncovering the etiology of psychiatric problems[17]. Nevertheless, the primary goal of machine learning models is to identify brain biomarkers that can improve the diagnoses, treatment, and prevention of psychiatric disorders in the broad population, not only for one specific group. A more ideal approach could be similar to the risk calculator developed in medical research[67], a more standardized protocol in genetic association studies[68], or some prediction pipelines developed in non-psychiatric studies[25]. The common characteristic of these examples is that model weights ('gold-standard') are applied to different populations with diverse backgrounds, which creates a high demand for the model as well as the identification of common biomarkers. If model performance varied across some groups (e.g., sex, age, cultural backgrounds), the important predictors could be included in the extension and further validation of models to create a more generalizable, clinically useful model.

In our study, the lack of this gold-standard generalizability in an external, independent sample suggests limited external validity, meaning that the dimensions cannot be applied to other datasets as potential biomarkers in the general population. This is certainly a concerning observation, given the two cohorts utilized in this study represent the largest studies of neurodevelopment in the world and are uniquely positioned to conduct such multivariate analyses. In the subsequent paragraphs, we will delve into potential explanations for why this step remains so challenging for psychiatric neuroimaging as applied to the general population, and then provide recommendations on how to improve out-of-study generalizability.

First, the multivariate CCA method is highly prone to overfitting and instability[20,69] and requires a large sample size to obtain sufficient statistical power[52]. In our study, the sample size of Generation R (*n* = 2043) might not be large enough to capture the associations that we found in ABCD. Yet, Generation R is a larger sample compared with the ABCD test set (*n* ~ 1000), where we successfully validated the associations. Second,

focusing on the general population might dilute the associations. The vast majority of previous studies drew from clinical samples with specific diagnoses, such as major depression and psychosis[2,56]. Since healthy individuals are overrepresented in population-based samples, the effect sizes will likely be smaller than in clinical samples and may be more difficult capture[70]. However, the utility of a dimensional assessment of symptoms is well-known and has several advantages to problems in clinical, case-control designs, which are also prone to overfitting and bias[71]. Third, resting-state fMRI data has intrinsic high inter-individual variability and smaller effect sizes at the individual level than other brain measures in psychiatry[72]. Thus, extracting clinically important signals on an individual basis is difficult, and generalizability across cohorts could be especially challenging. This can be seen from our results: the psychopathology profiles were relatively stable within ABCD as well as across cohorts, but the brain phenotypes associated with the behavioral profiles were highly unstable.

Another possible explanation is that brain-behavior associations differ across populations and cultures due to unconsidered confounders. Model failures are usually interweaved with other factors[7], such as differences in reporting preference and symptom presentation in diverse populations, which may correspond to divergent neurobiological underpinnings. The internally valid associations in ABCD could be cohort-specific effects that are not entirely consistent with Generation R. Although the eight-syndrome structure of CBCL was shown to be stable across different societies[73,74], our results could reflect, to some extent, the different brain-symptom construction across cultures.

Given these challenges, we recommend that future studies test results in a fully external validation dataset from a different population/cohort using the gold-standard generalizability test. Leave-site-out cross-validation can be an alternative if external validation in another study is not possible in practice, although this is not optimal as the true out-of-study generalizability. Moreover, except for data harmonization, hidden confounders across sites or studies should be considered. This could be discerned by assessing the

distribution of important potential confounders across sites or studies, and decisions could be made on which models and predictors to use accordingly. Recent advances in methods of accommodating site variations might also considerably boost generalizability and improve the site differences[75].

## Limitations

Using the largest multicohort study investigating the multivariate brain-behavior associations in pre-adolescence, the enhanced statistical power allowed us to examine whether robust associations can be detected and generalized in the general population. Despite the strengths, a few limitations should be noted. First, we only applied SCCA in our analysis, other doubly multivariate methods, such as Partial Least Squares (PLS), were not examined. However, CCA is one of the most widely used techniques, and other multivariate methods have been found to be sensitive to similar problems of generalizability[5,21,53]. Second, the conclusions drawn from the current study might not generalize to clinical samples. Although similar poor out-of-sample multivariate associations were seen in clinical samples[21], prediction models built in clinical studies might be more robust due to potentially larger effects. Yet, biomarkers emerging from the general population are useful in screening high-risk individuals, prevention, and health education, which are also important in health care practices. Third, the limited generalizability in the current study came from the failure to generalize in one specific dataset. We did not comprehensively test multiple datasets. Fourth, we did not investigate the specific causes underlying the difference between different sites and cohorts, which should be further investigated in future studies.

## Conclusions

In summary, the utilization of SCCA enabled us to discover robust brain-symptom associations in ABCD but limited external validity in Generation R. Overall, the results offer substantial room for optimism about using multivariate methods in brain-behavior association studies. Nonetheless, we strongly suggest that future studies test the generalizability of results in a fully external validation dataset using gold-standard tests. In this way, we are able to provide insights into how the results vary across contexts and populations and evaluate the full picture of model performance. To achieve the ultimate goal of clinical utility, transparently reporting and sharing the analysis workflows, collaborations across labs, and an open discussion about how to define good external validity can facilitate the robustness of scientific endeavors in brain-behavior association studies.

## Data availability

The ABCD data reported in this paper are openly available upon approval from the NDA Data Access Committee (https://nda.nih.gov/). The ABCD data came from ABCD collection 3165 (ABCD-BIDS Community Collection (ABCC), https://collection3165.readthedocs.io) and the Annual Release 4.0 (10.15154/1523041). According to local, national, and European Union regulations and the informed consent status of the participants, the Generation R datasets may be made available upon request to the Director of the Generation R Study, Vincent Jaddoe (v.jaddoe@erasmusmc.nl). The group-level numerical data underlying figures and tables in the manuscript are publicly available in the following Open Science Framework (OSF) repository: https://osf.io/8e3nf/.

## Code availability

All analysis code is publicly available in the following GitHub repository: (https://github.com/EstellaHsu/Brain_dimensions_ABCD_GenR), https://doi.org/10.5281/zenodo.10513071.

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

## Acknowledgements

Data used in the preparation of this article were obtained from the Adolescent Brain Cognitive Development[SM] (ABCD) Study (https://abcdstudy.org), held in the NIMH Data Archive (NDA). This is a multisite, longitudinal study designed to recruit more than 10,000 children age 9–10 and follow them over 10 years into early adulthood. The ABCD Study® is supported by the National Institutes of Health and additional federal partners under award numbers U01DA041048, U01DA050989, U01DA051016, U01DA041022, U01DA051018, U01DA051037, U01DA050987, U01DA041174, U01DA041106, U01DA041117, U01DA041028, U01DA041134, U01DA050988, U01DA051039, U01DA041156, U01DA041025, U01DA041120, U01DA051038, U01DA041148, U01DA041093, U01DA041089, U24DA041123, U24DA041147. A full list of supporters is available at https://abcdstudy.org/federal-partners.html. A listing of participating sites and a complete listing of the study investigators can be found at https://abcdstudy.org/consortium_members/. ABCD consortium investigators designed and implemented the study and/or provided data but did not necessarily participate in the analysis or writing of this report. This manuscript reflects the views of the authors and may not reflect the opinions or views of the NIH or ABCD consortium investigators. The Generation R Study is supported by Erasmus MC, Erasmus University Rotterdam, the Rotterdam Homecare Foundation, the Municipal Health Service Rotterdam area, the Stichting Trombosedienst & Artsenlaboratorium Rijnmond, the Netherlands Organization for Health Research and Development (ZonMw), and the Ministry of Health, Welfare and Sport. Neuroimaging data acquisition was funded by the European Community's 7th Framework Program (FP7/2008-2013, 212652, Nutrimenthe). Netherlands Organization for Scientific Research (Exacte Wetenschappen) and SURFsara (Snellius Compute Cluster, www.surfsara.nl) supported the Supercomputing resources. Authors are supported by an NWO-VICI grant (NWO-ZonMW: 016.VICI.170.200 to H.T.) for H.T., B.X., and the Sophia Foundation S18-20, and Erasmus MC Fellowship for R.L.M.

## Author contributions

Based on the CRediT role taxonomy (https://credit.niso.org/): conceptualization (B.X., J.F., H.T., and R.L.M.), data collection (B.X., L.D.A., P.C., M.L., H.T. and R.L.M.), data curation (B.X., P.C., M.L., and R.L.M.), formal analysis (B.X.), funding acquisition (H.T. and R.L.M.), investigation (B.X.), methodology (B.X., G.B., H.W., H.T. and R.L.M.), project administration (B.X., H.T., and R.L.M.), resources (R.L.M.), software (B.X. and L.D.A.), supervision (H.T. and R.L.M.), validation (B.X.), visualization (B.X.), writing—original draft (B.X.), writing—review and editing (B.X., L.D.A., J.F., G.B., B.T.C., P.C., M.B., M.L., A.M., H.W., H.T. and R.L.M.).

## Competing interests

The authors declare no competing interests.
