## [Peer Review File · Communications Psychology]

13th Jun 23

Dear Professor Tiemeier,

Thank you for your patience during the peer-review process. Your manuscript titled "Multivariate brain-based dimensions of child psychiatric problems: degrees of generalizability" has now been seen by 4 reviewers, whose comments are appended below. You will see that they find your work of some potential interest. However, they have raised quite substantial concerns that must be addressed. In light of these comments, we cannot accept the manuscript for publication, but would be interested in considering a revised version that fully addresses these serious concerns.

We hope you will find the Reviewers' comments useful as you decide how to proceed. Should additional work allow you to address these criticisms, we would be happy to look at a substantially revised manuscript. If you choose to take up this option, please highlight all changes in the manuscript text file, and provide a detailed point-by-point reply to the reviewers.

The three reviewers fall into two categories: technical concerns and the quality of presentation. In revision, we ask that you perform all necessary further analyses to alleviate the referees' concerns regarding the strength of evidence for your conclusion. In terms of presentation, please ensure that the existing literature is well-integrated and discussed in sufficient detail; that remaining ambiguities and caveats, such as the concerns about how low within-sample correlation affects across-sample generalization are discussed in a section titled "limitations" in the Discussion; and finally, that the presentation of methods and results is clear and also includes the information from the reporting summary.

If the revision process takes significantly longer than five months, we will be happy to reconsider your paper at a later date, provided it still presents a significant contribution to the literature at that stage.

Please use the following link to submit your revised manuscript, point-by-point response to the Reviewers' comments with a list of your changes to the manuscript text (which should be in a separate document to any cover letter) and any completed checklist:

[link redacted]

Please do not hesitate to contact me if you have any questions or would like to discuss the required revisions further. Thank you for the opportunity to review your work.

Best regards,

Antonia Eisenkoeck

Antonia Eisenkoeck
Senior Editor
Communications Psychology

EDITORIAL POLICIES AND FORMATTING

Editorial Policy: [Policy requirements](https://www.nature.com/documents/nr-editorial-policy-checklist.pdf) (Download the link to your computer as a PDF.)

Furthermore, please align your manuscript with our format requirements, which are summarized on the following checklist:

[Communications Psychology formatting checklist](https://www.nature.com/documents/commspsychol-style-formatting-checklist-article-rr.pdf)

and also in our style and formatting guide [Communications Psychology formatting guide](https://www.nature.com/documents/commspsychol-style-formatting-guide-accept.pdf) .

* **CODE AVAILABILITY:** All Communications Psychology manuscripts must include a section titled "Code Availability" at the end of the methods section. In the event of publication, we require that the custom analysis code supporting your conclusions is made available in a publicly accessible repository; please choose a repository that provides a DOI for the code; the link to the repository and the DOI must be included in the Code Availability statement. Publication as Supplementary Information will not suffice. We ask you to prepare and upload code at this stage, to avoid delays later on in the process.

*** DATA AVAILABILITY:**

All Communications Psychology research manuscripts must include a section titled "Data Availability" at the end of the Methods section or main text (if no Methods). More information on this policy, is available at <http://www.nature.com/authors/policies/data/data-availability-statements-data-citations.pdf>.

At a minimum the Data availability statement must explain how the data can be obtained and whether there are any restrictions on data sharing. Communications Psychology strongly endorses open sharing of data. If you do make your data openly available, please include in the statement:

We recommend submitting the data to discipline-specific, community-recognized repositories, where possible and a list of recommended repositories is provided at <http://www.nature.com/sdata/policies/repositories>.

If a community resource is unavailable, data can be submitted to generalist repositories such as [figshare](https://figshare.com/) or [Dryad Digital Repository](http://datadryad.org/). Please provide a unique identifier for the data (for example a DOI or a permanent URL) in the data availability statement, if possible. If the repository does not provide identifiers, we encourage authors to supply the search terms that will return the data. For data that have been obtained from publicly available sources, please provide a URL and the specific data product name in the data availability statement. Data with a DOI should be further cited in the methods reference section.

REVIEWER EXPERTISE:

Reviewer #1: RS fMRI connectivity, child psychiatry

Reviewer #2: multivariate machine learning

Reviewer #3: RS fMRI connectivity, child psychiatry

Reviewer #1 (Remarks to the Author):

The authors have attempted to delineate reproducible brain functional connectivity profiles that are associated with psychiatric problems in 10-11 year old children, across two large population-based cohorts. They employed sparse canonical correlations analysis and examined out-of-subsample and out-of-study generalizability. Modest brain-behavior associations were discovered in the ABCD data for three connectivity profiles that could be linked to meaningful CBCL correlates. However, out-of-

sample generalizability demonstrated a substantially weaker association in the test set compared to training set, and no out-of-study generalizability was observed. Major strengths of the work include the impressive source datasets, multivariate statistical approach and use of multiple approaches to generalizability. Nonetheless, there are several limitations to the approach that may have negatively impacted the results and interpretability of findings, as outlined below.

1. The introduction is focused rather heavily on psychiatric disorders and does not provide a strong rationale for investigating brain-behavior associations with CBCL subscales in pre-adolescence. The rationale for using a dimensional approach to psychopathology needs to be elaborated on. Here, the neurally informed structuring of psychopathology dimensions seems more relevant than diagnostic classification.

2. Introduction line 77-78 reads: “psychiatric neuroimaging studies have not generally adopted these external validation strategies” – this statement requires adequate referencing.

3. The authors acknowledge that the fMRI pre-processing steps differed between the ABCD and Generation R cohorts. Can the authors comment more extensively on how this impacts out-of-study generalizability? The study would be strengthened if the authors could demonstrate that the findings are robust against the varying pre-processing methods, for example by comparing both approaches in a smaller subset of the data.

4. Principal Component Analysis

- The authors selected the first 100 principal components in the training dataset to protect against overfitting. Based on the loss in effect size between the training and test datasets, some overfitting has occurred. Would it be worthwhile to repeat the analysis with a smaller set of components, to try and avoid overfitting?
- In lines 128-129 the authors describe out-of-sample generalisability as follows: “To reduce sampling biases, the split procedure was repeated 10 times, resulting in 10 pairs of independent train-test sets. Importantly, the analyses in ABCDTraining and ABCDTest sets were fully separated to safeguard the results from data leakage”. Could the principal components analysis (and other pre-processing steps) result in data leakage between the test and training sets within the ABCD study? Given the relatively large sample size of ABCD relative to Generation R, could the ABCD training set not have been split in a nested cross-validation approach, whereby the 10 splits are isolated to the ABCD training set and therefore the ABCD test set remains fully independent of training?
- How much variance did the 100 principal components explain in the training and test datasets?
- The authors need to clarify how identified components were labelled as networks. How similar were the identified components between the ABCD and Generation R studies? Could differences in breakdown of networks have contributed to the lack of generalizability observed across studies?
- The most influential components in figure 4b are not labelled as networks. Did the authors confirm that these components do not reflect noise?

5. CBCL subscales

- Is the number of CBCL items correct (118 instead of 113)? Could the authors state the CBCL version used from each study?
- Can the authors please more clearly describe the calculation of subscale total scores in the methods (including the range), provide a motivation for using raw total scores rather than T total scores, and add subscale descriptives for each study to Table 1? Reporting corresponding T-scores or percentiles would assist with comparison to previous literature related to brain-based dimensions of

CBCL syndrome scales.

- Mean internalizing and externalizing scores appear to be rather low, do the samples have sufficient coverage of clinically relevant symptoms for detecting brain-behavior dimensions associated with child psychiatric symptoms? Can the authors comment on this explicitly in the results or discussion?
- Have the authors confirmed that correlations were not impacted by CBCL outliers and the skewed distribution of CBCL scores?

6. Generalizability

- The authors describe one of their analyses (line 212) as “more commonly used qualitative replication”. While this type of replication is common in traditional statistical frameworks, it is much less common for complex multivariate and data-driven approaches such as sparse canonical correlation analysis. I suggest that the authors rephrase. In sentences 299-301 it needs to be clarified that the replication analyses were not repeated with the same multivariate model as in ABCD.
- With the qualitative replication approach, Pearson correlations between CBCL canonical loadings of the ABCD and Generation R studies were relatively high. The comparison of loadings is presented in Figure 5. It would be very interesting if the authors could add more information about how comparable the brain canonical loadings were between both studies. It would for example be informative to show the Generation R equivalents of figures 3d-f.

Reviewer #2 (Remarks to the Author):

The authors use canonical correlation analyses to make functional connectomes to a validate questionnaire on youth mental health. Within a large dataset, ABCD, they find that these canonical factors generalize across train and test splits. With another large dataset, Generation R, they find that these canonical factors do not generalize across the cohorts. But they do find that the results replicate across cohorts. Strengths include the inclusion of two large datasets from different cultural backgrounds and two different forms of generalization. Limitations include some confusing language and over statement of results, which could also be put into better context with the existing literature.

I have signed this review for transparency and am happy to discuss these comments if they are unclear. - Dustin Scheinost

I think some additional details are needed to describe the prediction pipeline. What data is used in the SCCA? I believe is just the training data with inputs from the PCA (100 components run just of the training data) and CBCL. But this is not really detailed or shown in figure 1. Also is the data used to estimate the PCA's the same as used in the SCCA? Finally, it is unclear where the elastic nets come in. Did the authors use an elastic net to create a predictive model or is it that the authors use an elastic net style penalty in the SCCA?

Various part are worded a little strongly. For example, “in order to safeguard against overfitting ...”. PCA on the connectivity matrices alone would not safeguard against overfitting. It would reduce the dimension of the data, which can help minimize overfitting. In general, it would be good if the authors could be very specific with their language. That will help any readers not misunderstand their findings.

More than 10 splits might be helpful. There is a large variation in results for the first canonical factor ($r=0.09-0.17$).

The culture between the Netherlands and the US are very different. The authors hint at this, but I think they could do a better job of situating this in terms of generalization. For example, even within ABCD prediction performance across different demographic groups exist, suggesting that generalizing mental health measures across cultures is hard. Indeed, we know that different communities have differences in symptom presentation, as well as differing perspectives on mental health more broadly. In all it is not surprising that models might not generalize across cultures. In other words, a lack of generalization might not be due to overfitting, bias, or poor methods. It may in fact represent a true difference in brain behavior associations between cohorts. See for example Tejavibulya L, et al. Predicting the future of neuroimaging predictive models in mental health. *Mol Psychiatry*. 2022 Aug;27(8):3129-3137. doi: 10.1038/s41380-022-01635-2. Epub 2022 Jun 13. PMID: 35697759; PMCID: PMC9708554.

This is especially true given the small effect sizes in the study. A correlation of $<.1$ does not have much room to lose any explained variance. For example, if the effects were around $r=.5$, losing half of the effect would still lead to a significant result. But with $r=.1$ losing half of the effect is likely to be insignificant.

Relatedly, the authors make the point that different results might be seen in clinical samples and that these studies largely draw from health individuals. This is an important point as the authors also note that for clinical psychiatric care, we would a biomarker that generalizes well. But the authors do not test in a clinical group. So some of the writing about how these results impact mental health research might be a bit over interpreted.

Did the authors try training in Generation R and testing in ABCD? It would be worth knowing if the same pattern of generalization (or lack thereof) is observed.

It would also be good to explicitly define terms like generalization and replication. While the terms are often used interchangeability. They can be different things to different researchers.

Reviewer #3 (Remarks to the Author):

In this study the authors assess the external cross validation of data-driven child psychopathology from rsfMRI connectivity and CBCL by utilizing the ABCD study baseline and the Generation R study samples. The question is important and merits scrutiny and publications from different groups, even if other work has already alluded to this issue.

1) The study's scope is presented in a manner that utilizing only one method does not seem comprehensive enough and other common methods than SCCA should be assessed and included somewhere, potentially in the supplementary section. Maybe at least one from the Kernel family (kernel ridge regression or support vector regression), as well as connectome-based predictive modeling, since these are used widely. Currently, only small variations in the SCCA are shown in the supplementary section.

2) The correlations are not particularly high even in the internal cross-validation (e.g. $r = .13$ for the

primary LV), which is consistent with other ABCD-based studies showing that, other than the cognitive domain, measures in the personality and mental health domains have low brain-phenotype associations (e.g. Chen, J., Tam, A., Kebets, V. et al. Shared and unique brain network features predict cognitive, personality, and mental health scores in the ABCD study. *Nat Commun* 13, 2217 (2022). <https://doi.org/10.1038/s41467-022-29766-8>). Given this, it is expected that such already small out-of-sample r would not survive an external cross-validation. I am wondering what the upper-bound of the cross-dataset validation is based on just the reliability of the psychopathology latent variables between ABCD and Gen R (i.e., ignoring brain)?

3) line 78: I would also cite previous work recommending these such as e.g.: Scheinost, D., Noble, S., Horien, C., Greene, A. S., Lake, E. M., Salehi, M., ... & Constable, R. T. (2019). Ten simple rules for predictive modeling of individual differences in neuroimaging. *NeuroImage*, 193, 35-45). Additionally, I would include some studies from non-psychiatric neuroimaging domain that involve external cross-validation (in addition to internal) after “medical research” and before the “psychiatric neuroimaging” to both temper the sentence and make the gap clearer (e.g. Avery, E. W., Yoo, K., Rosenberg, M. D., Greene, A. S., Gao, S., Na, D. L., ... & Chun, M. M. (2020). Distributed patterns of functional connectivity predict working memory performance in novel healthy and memory-impaired individuals. *Journal of cognitive neuroscience*, 32(2), 241-255.; Kardan, O., Stier, A. J., Cardenas-Iniguez, C., Schertz, K. E., Pruin, J. C., Deng, Y., ... & Rosenberg, M. D. (2022). Differences in the functional brain architecture of sustained attention and working memory in youth and adults. *Plos Biology*, 20(12), e3001938.

4) line 97: The sample size from ABCD in this study is much larger than other studies with adequate exclusion of head motion using ABCD rsfMRI (e.g. Wang, Z., Zhou, X., Gui, Y. et al. Multiple measurement analysis of resting-state fMRI for ADHD classification in adolescent brain from the ABCD study. *Transl Psychiatry* 13, 45 (2023). <https://doi.org/10.1038/s41398-023-02309-5> or Sripada, C., Rutherford, S., Angstadt, M. et al. Prediction of neurocognition in youth from resting state fMRI. *Mol Psychiatry* 25, 3413–3421 (2020). <https://doi.org/10.1038/s41380-019-0481-6>). Please elaborate on this in the methods or discussion.

Reviewer #1

The authors have attempted to delineate reproducible brain functional connectivity profiles that are associated with psychiatric problems in 10-11-year-old children, across two large population-based cohorts. They employed sparse canonical correlations analysis and examined out-of-subsample and out-of-study generalizability. Modest brain-behavior associations were discovered in the ABCD data for three connectivity profiles that could be linked to meaningful CBCL correlates. However, out-of-sample generalizability demonstrated a substantially weaker association in the test set compared to training set, and no out-of-study generalizability was observed. Major strengths of the work include the impressive source datasets, multivariate statistical approach and use of multiple approaches to generalizability. Nonetheless, there are several limitations to the approach that may have negatively impacted the results and interpretability of findings, as outlined below.

We are grateful to the reviewer for the valuable comments and suggestions.

(1). The background information: the introduction is focused rather heavily on psychiatric disorders and does not provide a strong rationale for investigating brain-behavior associations with CBCL subscales in pre-adolescence. The rationale for using a dimensional approach to psychopathology needs to be elaborated on. Here, the neurally informed structuring of psychopathology dimensions seems more relevant than diagnostic classification.

(1). We thank the Reviewer for this important comment and adapted the Introduction accordingly:

Introduction (page 4):

“This approach aims to identify neurally informed dimensions of psychopathology in the general population, transcending different domains of psychiatric problems and including the continuum of symptoms. This is complementary to approaches using diagnostic categorizations in clinical samples, which has faced challenges due to high heterogeneity within a given disorder and comorbidity across disorders¹. Moreover, subthreshold cases which are close to but do not meet diagnostic criteria are also important for understanding psychiatric disorders², but are not considered in dichotomization approaches. This is especially problematic for children, whose psychiatric disorders are widely recognized as dimensional³. The current study, therefore, adopted the dimensional approach in the general population in order to delineate novel neurobiological structures of child psychiatric problems.”

Introduction (page 6):

“By leveraging two large population-based samples, we were able to capture the continuum of psychiatric symptoms transdiagnostically. This enabled us to depict the brain-based dimensions of child psychopathology.”

(2). Introduction line 77-78 reads: “psychiatric neuroimaging studies have not generally adopted these external validation strategies” – this statement requires adequate referencing.

(2). We have referenced related studies and revised the Introduction to make the study gap more precise:

Introduction (page 5):

“One of the key elements that is largely missing from previous work is robust external validation in a fully independent dataset (i.e., not a held-out subsample from the same dataset).

This has been widely implemented in the validation of prediction models in medical research^{4,5} and recommended as a necessary step in prediction models⁶. While several non-psychiatric neuroimaging studies have established more standardized analysis pipelines⁷⁻⁹, most multivariate psychiatric neuroimaging studies have not generally adopted these stringent external validation strategies¹⁰⁻¹⁶.”

(3). The authors acknowledge that the fMRI pre-processing steps differed between the ABCD and Generation R cohorts. Can the authors comment more extensively on how this impacts out-of-study generalizability? The study would be strengthened if the authors could demonstrate that the findings are robust against the varying pre-processing methods, for example by comparing both approaches in a smaller subset of the data.

(3). We thank the Reviewer for raising this very important point. In the original manuscript, the preprocessing pipelines were different across cohorts (ABCD used the HCP pipeline, and Generation R used the fMRIPrep pipeline). We agree that the differences in preprocessing in the two cohorts could be concerning for the poor out-of-study generalizability. To address this concern, we extracted the recently available resting-state time data from the ABCD-BIDS Community Collection (ABCC), which underwent the **same preprocessing** procedure as Generation R (fMRIPrep preprocessing pipeline)¹⁷. We calculated the functional connectivity matrices in both cohorts using the same standard space template, time series confounder regressors, and brain atlas, and then re-ran our SCCA analysis pipeline. The analyses now represent a **highly harmonized** situation of the resting-state data across two independent cohorts. We updated the Methods, Results, and Discussion accordingly. Generally, the results remained largely the same.

Methods (page 20):

“fMRI pre-processing

The BIDS data were preprocessed with the fMRIPrep pipeline¹⁷ both in ABCC (version 20.2.0) and GenR (version 20.2.7). Briefly, anatomical MRI data first underwent intensity normalization to account for B₁-inhomogeneity and brain extraction, followed by nonlinear registration to MNI space and FreeSurfer processing. Functional MRI data underwent volume realignment with MCFLIRT (FSL). BOLD runs were then slice-time corrected with 3dTshift (AFNI), followed by co-registration to the corresponding T1w reference. Data were ultimately resampled to FreeSurfer fsaverage5 surface space. Of note, only the first run of resting-state data in ABCD was extracted to further optimize comparability with GenR.

Parcellation and whole-brain connectivity estimation

The connectivity estimation procedure was identical in ABCD and GenR. Briefly, whole-brain functional connectivity matrices were calculated using the Gordon cortical parcels¹⁸ and FreeSurfer subcortical segmentation¹⁹, yielding 349 distinct parcels consisting of 333 cortical and 16 subcortical regions. Briefly, after removing the first 4 volumes from each dataset to ensure magnetic stability, the BOLD signals were averaged across all voxels in each cortical and subcortical region. Then the extracted time series were adjusted for CSF and white matter signals (plus their temporal derivatives and quadratic terms), low-frequency temporal regressors for high-pass temporal filtering, and 24 motion regressors (6 base motion parameters + 6 temporal derivatives + 12 quadratic terms). Pearson correlation was applied to estimate the temporal dependence between the residualized regional time series, and the estimated

connectivity was Fisher z-transformed, resulting in a symmetric 349×349 functional connectivity matrix for each participant.”

The updated processing pipeline not surprisingly led to changes throughout the Results Section. Though these changes are broad as many specific values (correlation coefficients, p-values, etc.) have changed, the general pattern of results remained highly similar. Below are some selected examples of the changes:

Results (page 8):

“Overall, we found evidence that the first canonical correlation was robustly identified across train-test splits, the second to a lesser extent, and the third failed to be validated, in the ABCD_{Test} sets. Specifically, the first dimension was validated across 25 out of 30 splits ($r_1 = 0.12$, $ps < .05$ from permutation testing, **Supplementary Table 3**). This brain-symptom dimension captured the correlates between attention problems and connectivity patterns in connectivity networks involved in higher-order functions (salience, cingulo-opercular, and frontoparietal network)²⁰, visual-spatial attention network (parietal occipital, medial parietal)²¹, and motor networks (**Figure 3a, 3c**). The second dimension was only evident in 13 out of 30 splits ($r_2 = 0.07$, $ps < .05$ from permutation testing). This linked dimension delineated a relationship between rule-breaking and aggressive behaviors and connectivity patterns in similar networks involved in higher-order and visual-spatial attention functions, with a larger contribution from subcortical areas and motor networks (**Figure 3b, 3d**). Interestingly, across two linked dimensions, the salience, parietal occipital, motor, and cingulo-opercular networks, were overlapped. The third dimension was only observed in 4 of the 30 splits ($r_3 = 0.05$, $ps > .05$ from permutation testing; all three presented correlations were averaged across 30 splits, and p-values were corrected for multiple testing using the False Discovery Rate), thus was not considered a stable, internally valid dimension.”

Results (page 10):

“In the gold-standard generalizability test, we only observed the first canonical correlations survived permutation testing in 1 of the 30 train-test splits (**Supplementary Table 3**). All other canonical correlations did not survive in GenR when we used the SCCA models trained in ABCD ($r_1 = 0.03$, $r_2 = 0.03$, $r_3 = 0.02$, $ps > 0.05$; correlations averaged across 30 train-test splits, **Table 3, Figure 2b**).”

Results (page 11):

“After the permutation test, four significant canonical variates were identified in Generation R. Specifically, three similar canonical correlations were also observed in Generation R (**Figure 5a, 5b**), showing a Pearson correlation of $r = 0.87$ in the CBCL canonical loadings of attention problems, $r = 0.46$ in aggressive and rule-breaking behaviors, and $r = 0.31$ in anxious and withdrawn behaviors.”

Discussion (page 14):

“In the present study, we observed highly similar behavioral dimensions when training the SCCA model independently in Generation R. The robust dimensions observed in the discovery set (ABCD), alongside the similar behavioral dimensions observed in the qualitative replication, lend support for the reasonable internal validity of the brain-behavior dimensions.”

Table 3*Failed gold-standard generalizability test in Generation R*

Canonical correlations	ABCD		Generation R
	training set	test set	
r_1	0.17	0.12*	0.03
r_2	0.16	0.07*	0.03
r_3	0.14	0.05	0.02

Note. Canonical correlations in ABCD were averaged across the 30 train-test splits.

r_1 : * $p < 0.05$ in 25 train-test splits. r_2 : * $p < 0.05$ in 13 train-test splits.

r_1 Generation R: $p < 0.01$ in 1 train-test splits.

Supplementary table 3*Canonical correlations in ABCD and Generation R (10 splits as an example)*

		ABCD (n = 4,892)				Generation R n = 2,043
	Canonical Correlations	Training	Test	Sparsity		
Split 1		n=4,255	n=637			
	r_1	0.23	0.13***	rs-fMRI	0.6	0.02
	r_2	0.23	0.01	CBCL	0.5	0.03
	r_3	0.19	0.08			0.02
Split 2		n=4,092	n=800			
	r_1	0.23	0.13***	rs-fMRI	0.7	0.02
	r_2	0.23	0.09*	CBCL	0.5	0.01
	r_3	0.21	0.10**			0.002
Split 3		n=4,295	n=597			
	r_1	0.23	0.08	rs-fMRI	0.6	0.09
	r_2	0.22	0.07	CBCL	0.5	0.04
	r_3	0.21	0.01			0.01
Split 4		n=3,976	n=916			
	r_1	0.24	0.15***	rs-fMRI	0.9	0.02
	r_2	0.24	0.09*	CBCL	0.5	0.01
	r_3	0.22	0.03			0.001
Split 5		n=4,167	n=725			
	r_1	0.24	0.11*	rs-fMRI	0.8	0.03
	r_2	0.24	0.02	CBCL	0.5	0.04
	r_3	0.21	0.07			0.003

		n =4,401	n =491			
Split 6	r ₁	0.20	0.07	rs-fMRI	0.4	0.02
	r ₂	0.19	0.07	CBCL	0.5	0.001
	r ₃	0.18	0.06			0.03
		n =4,229	n =663			
Split 7	r ₁	0.23	0.14***	rs-fMRI	0.6	0.01
	r ₂	0.22	0.12**	CBCL	0.5	0.02
	r ₃	0.20	0.05			0.02
		n =4,139	n =753			
Split 8	r ₁	0.22	0.16***	rs-fMRI	0.6	0.03
	r ₂	0.21	0.10**	CBCL	0.5	0.02
	r ₃	0.21	0.04			0.02
		n =3,811	n =1,081			
Split 9	r ₁	0.23	0.19***	rs-fMRI	0.8	0.01
	r ₂	0.24	0.16***	CBCL	0.5	0.01
	r ₃	0.21	0.08*			0.01
		n =4,119	n =773			
Split 10	r ₁	0.24	0.10*	rs-fMRI	0.5	0.01
	r ₂	0.23	0.08*	CBCL	0.5	0.02
	r ₃	0.19	0.04			0.01

Note. The significance canonical correlations was assessed by permutation tests.

P values were corrected for multiple testing by False Discovery Rate (FDR).

* $p < 0.05$, ** $p < 0.01$, *** $p < 0.001$

Figure 2

Associated dimensions of brain connectivity and CBCL syndrome scores in ABCD

Note. SCCA identified brain-behavior correlates in training and test sets of ABCD. **a.** The first three canonical correlations were significant in the ABCD_{Training} sets. The canonical loadings of CBCL syndrome scores in the ABCD_{Training} set were averaged across 30 train-test splits. **b.** The mean and standard deviation of the first three canonical correlations across 30 train-test splits. **c.** Covariance explained in the training and test sets (example from one train-test split). The number of canonical variates in the ABCD_{Training} set that was put into the permutation test was selected based on the mean covariance explained. **d.** The first canonical variate was largely generalizable in ABCD_{Test} set across the 30 train-test splits, the second to a less extent, and the third was not generalizable. The red dotted lines represent the canonical correlations in the unshuffled data.

Figure 3

Resting-state connectivity canonical variates in ABCD

Note. Brain connectivity modules involved in the two identified canonical variates in ABCD. The contribution of each connectivity feature was determined by computing the correlations between the original connectivity matrix and the canonical variate scores of the brain connectivity extracted from the SCCA model (calculated by canonical loadings averaged across 30 train-test splits and the whole sample of ABCD), indicating the importance of each connectivity feature. After calculating the contribution of each connectivity feature, we summarized the contributions based on pre-assigned network modules and calculated the within and between-network loadings based on the network module analysis method in Xia, et al. (2018). **a-b.** The top 20% of the connectivity patterns that contributed most for each of canonical variate. The outer labels represent the names of network modules. The thickness of the chords showed the importance of different network modules. **c-d.** The connectivity patterns associated with the first two canonical variates. This is based on the z-scores of the within- and between-network loadings we calculated.

Figure 4

Stability and sampling variability of canonical correlations and canonical loadings in ABCD (example)

Note. Sampling variability and important contributors for the first three canonical variates. **a.** The variability for the canonical loadings of CBCL syndrome scores across 1000 bootstrap subsamples. The center line is median, the upper quantile is 75% and the lower quantile is 25%. **b.** The variability for the canonical loadings of brain PCs across 1000 bootstrap subsamples. The PCs presented here were selected based on the intersection of top 10 most important PCs for the first three canonical variates. **c.** The variability of the first three canonical correlations in ABCD_{Training} and ABCD_{Test} set. The black dot is mean, and the vertical black line is standard deviation. Note that the bootstrap subsampling is conducted in one of the 30 train-test splits. **CV1**: canonical variate 1, **CV2**: canonical variate 2, **CV3**: canonical variate 3.

Figure 5

CBCL canonical loadings in ABCD and Generation R in qualitative replication

Note. The comparison of canonical loadings for CBCL syndrome scores in ABCD and Generation R. **a.** The canonical loadings of CBCL syndrome scores in ABCD. **b.** The canonical loadings of CBCL syndrome scores in Generation R. **CV1**: canonical variate 1, **CV2**: canonical variate 2, **CV3**: canonical variate 3, **CV4**: canonical variate 4.

(4). Principal Component Analysis

4a. The authors selected the first 100 principal components in the training dataset to protect against overfitting. Based on the loss in effect size between the training and test datasets, some overfitting has occurred. Would it be worthwhile to repeat the analysis with a smaller set of components, to try and avoid overfitting?

4b. How much variance did the 100 principal components explain in the training and test datasets?

(4a, 4b). We agree with the Reviewer’s suggestion and have implemented this, with adaptations to the Methods, Results, and Supplemental Information sections:

Results (page 7):

“Three brain-symptom dimensions (canonical variates) were identified from the functional connectivity data (100 principal components, with averaged explained variance of 61.9% across 30 train-test splits) and the psychiatric symptom data (raw sum scores of the 8 CBCL syndrome scales).”

Results (page 10):

“We projected the SCCA model weights of the ABCD_{Training} set onto the first 100 brain PCs (explained variance 61.8%) and CBCL syndrome scores of Generation R.”

The selection of the number of principal components in our study was based on previous studies²². We agree that reducing the number of PCs might mitigate overfitting. To address this point, we reran the analyses with different numbers of PCs (i.e., 50, 200) in 10 train-test splits in ABCD. Overall, the results remained similar to results from 100 brain PCs, suggesting that a smaller number of brain PCs did not change the poor out-of-study generalizability.

Methods (page 29):

“Fourth, we included different numbers of brain principal components to inspect possible fluctuation of results due to the dimensionality of brain data, which might be one of the reasons for overfitting.”

Results (page 10):

“These results are highly consistent when we put different numbers of brain PCs (i.e., 50 and 200) in the model (**Supplementary Table 7, Supplementary Table 8, Supplementary Figure 1**).”

Supplementary table 6

Canonical correlations in ABCD and Generation R (50 brain PCs)

		ABCD (n = 4,892)				Generation R n = 2,043
Canonical Correlations		Training	Test	Sparsity		
Split 1		n =4,255	n =637			
	r ₁	0.20	0.15***	rs-fMRI	0.8	0.05
	r ₂	0.18	0.08	CBCL	0.5	0.04
	r ₃	0.16	0.04			0.09*
Split 2		n =4,092	n =800			
	r ₁	0.20	0.13***	rs-fMRI	0.7	0.01
	r ₂	0.19	0.10**	CBCL	0.5	0.001
	r ₃	0.15	0.08			0.005
Split 3		n =4,295	n =597			
	r ₁	0.21	0.06	rs-fMRI	0.8	0.07*
	r ₂	0.19	0.03	CBCL	0.5	0.04
	r ₃	0.17	0.03			0.05
Split 4		n =3,976	n =916			
	r ₁	0.19	0.15***	rs-fMRI	0.9	0.02
	r ₂	0.20	0.11***	CBCL	0.5	0.001
	r ₃	0.16	0.07			0.003
Split 5		n =4,167	n =725			
	r ₁	0.21	0.13**	rs-fMRI	0.9	0.003
	r ₂	0.20	0.06	CBCL	0.5	0.007
	r ₃	0.16	0.08			0.002
Split 6		n =4,401	n =491			
	r ₁	0.20	0.09*	rs-fMRI	0.7	0.06*
	r ₂	0.19	0.09	CBCL	0.5	0.07*
	r ₃	0.17	0.08			0.08*
Split 7		n =4,229	n =663			
	r ₁	0.20	0.14***	rs-fMRI	0.7	0.01
	r ₂	0.19	0.15**	CBCL	0.5	0.02
	r ₃	0.17	0.07			0.02
Split 8		n =4,139	n =753			
	r ₁	0.20	0.17***	rs-fMRI	0.8	0.06*
	r ₂	0.19	0.10**	CBCL	0.5	0.06
	r ₃	0.16	0.05			0.03
Split 9		n =3,811	n =1,081			
	r ₁	0.19	0.20***	rs-fMRI	0.9	0.01
	r ₂	0.18	0.14***	CBCL	0.5	0.01
	r ₃	0.18	0.08**			0.04
Split 10		n =4,119	n =773			
	r ₁	0.20	0.15***	rs-fMRI	0.7	0.11***
	r ₂	0.18	0.11**	CBCL	0.5	0.06
	r ₃	0.17	0.09*			0.12***

Note. The significance canonical correlations was assessed by permutation tests.

P values were corrected for multiple testing by False Discovery Rate (FDR).

* $p < 0.05$, ** $p < 0.01$, *** $p < 0.001$

Supplementary table 7

Canonical correlations in ABCD and Generation R (200 brain PCs)

		ABCD (n = 4,892)				Generation R n = 2,043
Canonical Correlations		Training	Test	Sparsity		
Split 1		n =4,255	n =637			
	r ₁	0.29	0.12***	rs-fMRI	0.8	0.03
	r ₂	0.27	0.05	CBCL	0.5	0.04
	r ₃	0.30	0.02			0.06
Split 2		n =4,092	n =800			
	r ₁	0.31	0.15***	rs-fMRI	0.8	0.02
	r ₂	0.28	0.08*	CBCL	0.5	0.06*
	r ₃	0.29	0.14***			0.004
Split 3		n =4,295	n =597			
	r ₁	0.29	0.08*	rs-fMRI	0.7	0.05
	r ₂	0.26	0.08	CBCL	0.5	0.05
	r ₃	0.29	0.04			0.03
Split 4		n =3,976	n =916			
	r ₁	0.29	0.16***	rs-fMRI	0.7	0.01
	r ₂	0.28	0.12**	CBCL	0.5	0.01
	r ₃	0.28	0.06			0.01
Split 5		n =4,167	n =725			
	r ₁	0.29	0.14***	rs-fMRI	0.8	0.003
	r ₂	0.30	0.07	CBCL	0.5	0.01
	r ₃	0.26	0.06			0.02
Split 6		n =4,401	n =491			
	r ₁	0.24	0.06	rs-fMRI	0.4	0.04
	r ₂	0.24	0.07	CBCL	0.5	0.07*
	r ₃	0.22	0.08			0.06
Split 7		n =4,229	n =663			
	r ₁	0.29	0.13**	rs-fMRI	0.7	0.06*
	r ₂	0.26	0.11**	CBCL	0.5	0.06*
	r ₃	0.30	0.06			0.04
Split 8		n =4,139	n =753			
	r ₁	0.26	0.19***	rs-fMRI	0.5	0.04
	r ₂	0.24	0.09**	CBCL	0.5	0.04
	r ₃	0.26	0.03			0.06*
Split 9		n =3,811	n =1,081			
	r ₁	0.29	0.18***	rs-fMRI	0.7	0.02
	r ₂	0.29	0.11**	CBCL	0.5	0.01
	r ₃	0.28	0.08*			0.001
Split 10		n =4,119	n =773			
	r ₁	0.30	0.11*	rs-fMRI	0.6	0.06*
	r ₂	0.26	0.11*	CBCL	0.5	0.06*
	r ₃	0.27	0.06			0.03

Note. The significance canonical correlations was assessed by permutation tests.

P values were corrected for multiple testing by False Discovery Rate (FDR).

* $p < 0.05$, ** $p < 0.01$, *** $p < 0.001$

Supplementary Figure 1

Canonical correlations and CBCL loadings across 10 splits (different brain PCs)

a. Canonical correlations across 10 splits (50 PCs)

b. CBCL canonical loadings across 10 splits (50 PCs)

c. Canonical correlations across 10 splits (100 PCs)

d. CBCL canonical loadings across 10 splits (100 PCs)

e. Canonical correlations across 10 splits (200 PCs)

f. CBCL canonical loadings across 10 splits (200 PCs)

Note. **a-b.** Canonical correlations and CBCL loadings when we put 50 brain PCs in the SCCA model. **c-d.** Canonical correlations and CBCL loadings when we put 100 brain PCs in the SCCA model. **e-f.** Canonical correlations and CBCL loadings when we put 200 brain PCs in the SCCA model.

4c. In lines 128-129 the authors describe out-of-sample generalisability as follows: “To reduce sampling biases, the split procedure was repeated 10 times, resulting in 10 pairs of independent train-test sets. Importantly, the analyses in ABCD_{Training} and ABCD_{Test} sets were fully separated to safeguard the results from data leakage”. Could the principal components analysis (and other pre-processing steps) result in data leakage between the test and training sets within the ABCD study? Given the relatively large sample size of ABCD relative to Generation R, could the ABCD training set not have been split in a nested cross-validation approach, whereby the 10 splits are isolated to the ABCD training set and therefore the ABCD test set remains fully independent of training?

(4c). We thank the Reviewer for bringing up this point that we hope to clarify here. First, the principal component analysis (PCA) is unlikely to lead to data leakage in our analysis pipeline. The PCA was applied to the ABCD_{Training} set, and the eigenvectors (the rotation or weight vectors) retrieved from ABCD_{Training} set were then projected to ABCD_{Test} set. In this way, we obtained the brain PCs in ABCD_{Test} sets. We did not implement the PCA to the whole ABCD data set, which means the ABCD_{Test} set was fully independent of the PCA implementation in the ABCD_{Training} set. This is now clarified in the manuscript as follows:

Methods (page 23):

“Importantly, all the analyses in ABCD_{Training} sets and ABCD_{Test} sets were fully separated to protect the results from data leakage (**Figure 1**), including the residualization of brain data, weighted PCA, and the SCCA model. Specifically, the residualization was separately done in ABCD_{Training} sets and ABCD_{Test} sets, and the weighted PCA was first implemented in ABCD_{Training} sets, then the PCA eigenvectors retrieved from ABCD_{Training} set were applied to ABCD_{Test} set to derive brain PCs. Next, the SCCA model was trained in ABCD_{Training} set, where the penalty parameters of the SCCA models were selected in 100 further random splits of training (80% of ABCD_{Training} set) and validation set (20% of ABCD_{Training} set). After fitting the model with the optimal penalty parameters in the ABCD_{Training} set, the *out-of-sample* model generalizability was evaluated by projecting the CBCL and brain PCs loadings obtained from the ABCD_{Training} set to the ABCD_{Test} set.”

Second, if we understand correctly, the Reviewer meant that we could just split one training set and a small fully independent held-out data set from ABCD and implement 10-fold cross-validation within that one training set. We did this procedure in our original attempts to implement the SCCA model, however, we realized that the **split of training and held-out data set could lead to sampling bias**, which can be clearly seen in **Figure 2b** and **Supplementary Table 3**: different splits shown quite different generalizability (e.g., split 6 and 8). This motivated our design of repeating the split 30 times, which was also suggested by previous literature²³, to inspect the possible sampling bias. Therefore, our analysis pipeline is not a typical nested cross-validation approach: the 30 train-test splits were not designed for tuning parameters or assessing out-of-sample performance in one held-out data set (that is why we did not present the average canonical correlations across the splits in **Supplementary Table 3**). Instead, it is a multiple held-out framework, where we could observe possible fluctuation of results in different training and held-out sample split. This is especially relevant when we split the training and held-out data set based on study sites as it would be highly biased if we examine the out-of-sample performance only in certain held-out study sites, even though they are randomly assigned. This is now made clearer in the text as follows:

Methods (page 23):

“To inspect potential sampling biases, the split procedure was repeated 30 times, resulting in 30 pairs of independent train-test sets. This is suggested as a multiple held-out framework, which can examine and reduce possible fluctuation of results in different training and held-out sample split^{23,24}.”

4d. The authors need to clarify how identified components were labeled as networks. How similar were the identified components between the ABCD and Generation R studies? Could differences in breakdown of networks have contributed to the lack of generalizability observed across studies?

(4d). To determine the importance of networks for each PC, we calculated the importance of each connectivity feature (60,726 vectorized connectivity) for each PC using the following equation:

$$\frac{w_{ij}^2}{\text{variance}(c_j)}$$

w_{ij} is the eigenvector (direction) for principal component i and connectivity feature j , and c_j is connectivity feature j . We then summarized each feature's importance into different networks (see the figures below). We found that the important networks contributing to the first PC were similar in ABCD and GenR but were different for other PCs across cohorts. However, the underlying distributions of the two data sets can be very similar despite their PCs being different. This characteristic of the data/PCA is important in the context of the SCCA. Specifically, it is not sensible to judge whether the SCCA model can be generalized across different studies by inspecting the similarity of PCs, as the **PCs are new data space with minimal information loss** rather than the original datasets.

Moreover, as evidence of this, we also tried to project the rotation/eigenvectors derived from ABCD to Generation R, which is a typical way to calculate the “same” PCs across two cohorts. However, the out-of-study generalizability was even worse (none of the associations were generalizable across all the train-test splits). This again indicated that the different PCs across studies are unlikely driving the results.

4e. The most influential components in figure 4b are not labelled as networks. Did the authors confirm that these components do not reflect noise?

(4e). We mapped two of the most influential components in **Figure 4b** (i.e., PC7 and PC12, see figure above). In ABCD, the Cingulo-opercular network and Auditory networks are the most important contributors to PC7, and subcortical areas is the important contributor to PC12. While we cannot rule out that any PC reflects noise, the possibility that these PCs **only** reflect noise is improbable. Our study design with stringent out-of-sample and out-of-study model performance tests can mitigate the possibility that the results are based on noise, which can be further supported by the consistent signals (i.e., the first brain-behavior dimension for attention problems) observed across train-test splits in ABCD. In fMRI studies, the boundary between noise and signals is not always clear. Noise fluctuations could be due to many reasons across individual subjects (e.g., motion, physiological fluctuations, system-related instabilities)²⁵ and it is unlikely that SCCA can consistently predict the same noise coming from different sources in different resamples.

(5). CBCL subscales

5a. Is the number of CBCL items correct (118 instead of 113)? Could the authors state the CBCL version used from each study?

(5a). We thank the Reviewer for spotting this mistake. The number of CBCL items should be 113. We have updated the Methods section accordingly:

Methods (page 20):

“Child psychiatric symptoms were assessed using the Child Behavioral Checklist (school-age version) in both cohorts. The CBCL is a 113-item caregiver report with eight syndrome scales.”

5b. Can the authors please more clearly describe the calculation of subscale total scores in the methods (including the range), provide motivation for using raw total scores rather than T total scores, and add subscale descriptives for each study to Table 1? Reporting corresponding T-scores or percentiles would assist with comparison to previous literature related to brain-based dimensions of CBCL syndrome scales.

(5b). More detailed information about the subscale total scores of the CBCL and the descriptive statistics were provided in the manuscript. We relied on CBCL raw scores rather than T-scores because the authors of the instrument (Achenbach and Rescorla) explicitly state within the CBCL manual that raw scale scores should be used in statistical analysis in order to take account for the full range of variation in these scales (2001, p. 89)²⁶. Thurber & Sheehan (2012) describe this issue in depth, noting that “*the T transformation results in the elimination of the bottom part of the score distribution; lower scores at the mean and below are all assigned T scores of 50, not just the mean raw score as in non-truncated T scores. Hence, T scores truncate or reduce the range of variation*”²⁷. Furthermore, the normative sample for the CBCL is decades old, so T scores may not be appropriate for the current samples due to cohort effects. In addition, the normative sample from which the T-scores were derived is smaller in size than the ABCD and GenR samples.

Methods (page 20):

“The current analyses relied on raw scores from the CBCL, as is recommended by the instrument authors to preserve the full range of variation²⁶. Items belonging to a given syndrome scale were summed. In the case of missing items, if the missingness was less than 25%, a sum score was created accounting for missing items. Higher scores represent more problems. For the detailed statistics for the eight syndrome scales in ABCD and Generation R, see **Supplementary Table 9, Table 1.**”

Supplementary Table 9

Descriptive statistics of CBCL in ABCD and Generation R

CBCL syndrome scales	ABCD (n = 4,892)			Generation R (n = 2,043)		
	M(SD) (raw scores)	Range (raw scores)	M(SD) (T scores)	M(SD) (raw scores)	Range (raw scores)	M(SD) (T scores)
Anxious/depressed	2.6(3.1)	(0, 26)	53.5(6.1)	2.2(2.6)	(0, 19)	52.7(5.0)
Withdrawn/depressed	1.0(1.7)	(0, 14)	53.3(5.6)	1.1(1.6)	(0, 10)	53.7(5.6)
Somatic	1.5(1.9)	(0, 15)	54.9(6.0)	1.5(2.0)	(0, 15)	54.8(6.0)
Social	1.5(2.2)	(0, 18)	52.5(4.4)	1.5(2.1)	(0, 16)	52.5(4.3)
Aggressive	3.1(4.2)	(0, 36)	52.6(5.3)	2.7(3.5)	(0, 28)	52.0(4.3)
Rule-breaking	1.1(1.7)	(0, 18)	52.5(4.5)	0.9(1.4)	(0, 11)	52.0(3.6)
Thought problems	1.5(2.1)	(0, 18)	53.6(5.7)	1.5(2.0)	(0, 15)	53.4(5.5)
Attention problems	2.7(3.3)	(0, 20)	53.5(5.8)	3.0(3.0)	(0, 18)	53.6(5.0)
Internalizing problems	5.1(5.6)	(0, 51)	48.4(10.6)	4.7(5.0)	(0, 37)	48.1(9.9)
Externalizing problems	4.1(5.6)	(0, 48)	45.2(10.0)	3.6(4.6)	(0, 38)	44.4(9.2)

Table 1*Descriptive statistics of the discovery set (example) and the external validation set*

	Discovery set		External validation set	
	ABCD n = 4,892		Generation R n = 2,043	
	ABCD_{Training}	ABCD_{Test}		
N	4,230	662	N	2,043
Age (years), M(SD)	10.0 (0.6)	10.0 (0.6)	Age (years), M(SD)	10.1 (0.6)
Sex			Sex	
Girls, (%)	48.9	48.5	Girls, (%)	52.4
Race/ethnicity (%)			Nation of birth (%)	
White	58.3	48.8	Dutch	66.1
African American	12.0	11.2	Non-Dutch European	17.3
Hispanic	19.3	15.1	Non-European	16.6
Asian	1.5	5.9		
Others	8.9	18.0		
Parental education (%)			Maternal education (%)	
Low	4.8	5.1	Low	2.8
Medium	38.1	39.1	Medium	34.5
High	57.1	55.8	High	62.7
Child Behavior Checklist (CBCL), M(SD)			Child Behavior Checklist (CBCL), M(SD)	
Anxious/depressed	2.5(3.1)	2.8(3.3)	Anxious/depressed	2.2(2.6)
Withdrawn/depressed	1.0(1.6)	1.2(1.8)	Withdrawn/depressed	1.1(1.6)
Somatic	1.5(2.0)	1.6(1.9)	Somatic	1.5(1.9)
Social	1.5(2.1)	1.6(2.3)	Social	1.5(2.1)
Aggressive	3.0(4.2)	3.3(4.2)	Aggressive	2.7(3.5)
Rule-breaking	1.0(1.7)	1.2(1.9)	Rule-breaking	0.9(1.4)
Thought problems	1.5(2.1)	1.8(2.3)	Thought problems	1.5(2.0)
Attention problems	2.6(3.3)	3.0(3.4)	Attention problems	2.9(3.0)
Internalizing scores	5.0(5.5)	5.6(5.9)	Internalizing scores	4.7(5.0)
Externalizing scores	4.1(5.6)	4.5(5.7)	Externalizing scores	3.6(4.6)
Total scores	16.9(17.2)	19.1(18.0)	Total scores	16.6(15.1)

Note. Values are frequencies for categorical variables and means and standard deviations for continuous variables. The descriptive statistics for ABCD were based on one of the 30 train-test splits, other splits showed similar statistics.

M = Mean, SD = Standard Deviation

5c. Mean internalizing and externalizing scores appear to be rather low, do the samples have sufficient coverage of clinically relevant symptoms for detecting brain-behavior dimensions associated with child psychiatric symptoms? Can the authors comment on this explicitly in the results or discussion?

(5c). We thank the Reviewer for the comment. We acknowledge that we did not make the differences between the population-based samples included in this manuscript and the clinical samples clear. Population-based studies usually contain the full continuum of psychopathology in the general population. This means children with subthreshold, mild, and severe psychopathology were all included. About 6.9% of children in ABCD and 5.1% of children in GenR (based on the clinical cut-off of 93% quantile of CBCL total scores) have clinically relevant symptoms in our sample. We added this information to the Methods section. We have also now included text on this in the Discussion.

Methods (page 19):

“Accordingly, data from 4,892 participants, of which around 6.9% had clinically relevant total problem symptom scores, were available for analysis in ABCD.”

“2,043 participants (of which 5.1% had clinically relevant total problem symptom scores) were included in the final sample for analysis in GenR.”

Discussion (page 16):

“Second, focusing on the general population might dilute the associations. The majority of previous studies drew from clinical samples with a clinical diagnosis, such as major depression or psychosis^{13,28}. Since healthy individuals are overrepresented in population-based samples, the effect sizes will likely be smaller than in clinical samples and may be more difficult to capture.”

5d. Have the authors confirmed that correlations were not impacted by CBCL outliers and the skewed distribution of CBCL scores?

(5d). The possibility that the correlations were impacted by CBCL outliers and skewed distribution is small. First, the CBCL syndrome scores in our study are within the normal range of each subscale (**Supplementary Table 9 in comment #5b**) both in ABCD and GenR, thus we cannot describe points as outliers based on this. Second, SCCA is more robust to deviation from normality compared with traditional CCA²⁹. Third, our study design of different resample methods and non-parametric approach can make the results robust to the potential impact of outliers and skewness. Lastly, the large sample size of this study can improve the robustness of the estimates obtained in non-ideal conditions such as non-normality and skewness³⁰. We included text on this in the Methods section.

Methods (page 19):

“The raw sum scores of each syndrome scale were within the normal range both in ABCD and GenR.”

Methods (page 25):

“This method is more stable, more robust to deviation from normality, and does not have the main constraint of classic CCA: the number of observations should be larger than the number of variables¹¹.”

(6). Generalizability

6a. The authors describe one of their analyses (line 212) as “more commonly used qualitative replication”. While this type of replication is common in traditional statistical frameworks, it is much less common for complex multivariate and data-driven approaches such as sparse canonical correlation analysis. I suggest that the authors rephrase. In sentences 299-301 it needs to be clarified that the replication analyses were not repeated with the same multivariate model as in ABCD.

(6a). We thank the Reviewer for this critical comment and addressed the comment in the Results and Discussion sections. We agree that the implementation of qualitative replication is less common for other multivariate or data-driven methods in several fields. However, in studies in the field of psychiatry, this replication approach is more commonly used, and the interpretation of results remains subjective.

Results (page 10):

“In the other, more commonly used qualitative replication in doubly multivariate psychiatric studies^{11,12,31}, a new SCCA model was trained in Generation R, yielding another set of canonical loadings.”

Discussion (page 13):

“One approach consists of repeating the analysis pipeline and training a new model in data that were previously ‘unseen’ by the multivariate algorithm, and then correlating the model weights across studies. This is often referred as ‘replication’, in a test set from the same large participant pool or an external dataset. Similarities of behavioral or brain loadings are frequently used to indicate a successful replication^{11,12}.”

6b. With the qualitative replication approach, Pearson correlations between CBCL canonical loadings of the ABCD and Generation R studies were relatively high. The comparison of loadings is presented in Figure 5. It would be very interesting if the authors could add more information about how comparable the brain canonical loadings were between both studies. It would for example be informative to show the Generation R equivalents of figures 3d-f.

(6b). While the CBCL canonical loadings (i.e., attention problems, aggressive/rule-breaking behaviors, anxious/depressed) were highly similar in ABCD and GenR, the canonical correlations were not overly generalizable from ABCD to GenR. It can thus be expected that the similarities of the related networks were limited across cohorts. Indeed, **Supplementary Figure 2** shows that there was some overlap of the most important networks in ABCD and GenR. However, the contributions of other networks were not the same across the two cohorts. This also reveals the subjectivity of the interpretation of a qualitative replication: it is not defined what constitutes ‘successful’, ‘partly’, or failed replication. We added this information in the Results section.

Results (page 11):

“Several of the most important brain connectivity networks involved were overlapping between ABCD and GenR (**Supplementary Figure 2**). For instance, both in ABCD and GenR, the salience, parietal occipital, and motor networks were the most crucial contributors to the association of attention problems and brain connectivity. Similarly, the motor, auditory

networks, and subcortical areas contributed most to the correlates of aggressive/rule-breaking behaviors and brain connectivity. However, the contributions of other networks, or the collective effects of all the networks, were not entirely the same across the two cohorts, suggesting that the variability in the brain phenotypes is underlying the poor out-of-study generalizability.”

Supplementary Figure 2

Important brain connectivity networks in Generation R

Note. Important brain connectivity modules involved in the two identified canonical variates (attention problems, aggressive/rule-breaking behaviors), which are highly similar to the two identified in ABCD, in Generation R. **a-b.** The top 20% of the connectivity patterns that contributed most for each of canonical variate. The outer labels represent the names of network modules. The thickness of the chords showed the importance of different network modules. **c-d.** The connectivity patterns associated with the first two canonical variates. This is based on the z-scores of the within- and between-network loadings we calculated.

Reviewer #2

The authors use canonical correlation analyses to make functional connectomes to a validated questionnaire on youth mental health. Within a large dataset, ABCD, they find that these canonical factors generalize across train and test splits. With another large dataset, Generation R, they find that these canonical factors do not generalize across the cohorts. But they do find that the results replicate across cohorts. Strengths include the inclusion of two large datasets from different cultural backgrounds and two different forms of generalization. Limitations include some confusing language and overstatement of results, which could also be put into better context with the existing literature.

I have signed this review for transparency and am happy to discuss these comments if they are unclear. - Dustin Scheinost

We thank the Reviewer for the valuable comments and advice.

(1). I think some additional details are needed to describe the prediction pipeline. What data is used in the SCCA? I believe is just the training data with inputs from the PCA (100 components run just of the training data) and CBCL. But this is not really detailed or shown in figure 1. Also is the data used to estimate the PCA's the same as used in the SCCA? Finally, it is unclear where the elastic nets come in. Did the authors use an elastic net to create a predictive model or is it that the authors use an elastic net style penalty in the SCCA?

(1). We thank the Reviewer for these questions. The model was trained only in the ABCD_{Training} sets, so we input two datasets (per training-test split) into the SCCA model (*i*) the first 100 principal components of the vectorized resting-state connectivity data from ABCD_{Training} sets and (*ii*) the 8 CBCL syndrome scale scores from ABCD_{Training} sets. The input data used for PCA was the high-dimensional connectivity matrices from ABCD_{Training} sets, and the PCA was a dimensionality reduction step of the brain data prior to SCCA. The elastic nets were only used to determine the penalty parameters within the SCCA model²⁹. One penalty was applied to each matrix of brain PCs and CBCL scores. For ABCD_{Test}, as well as the out-of-study gold-standard test in GenR, we did not train the SCCA model. For the qualitative replication approach, we trained a new SCCA model in GenR. We modified **Figure 1** to make our methods clearer.

Figure 1
Analysis pipeline

Note. a-b. ABCD was the discovery set and Generation R as the external validation set. The discovery set was divided into training and test sets 30 times, resulting in 30 train-test pairs in ABCD. The eigenvectors of PCA from the ABCD_{Training} set were applied to ABCD_{Test} set to calculate the principal components, then the weight vectors (canonical loadings) obtained from the ABCD_{Training} set were projected to ABCD_{Test} set to compute the out-of-sample correlations. Similarly, weight vectors of SCCA from the ABCD_{Training} set were then directly applied to Generation R to assess the out-of-study generalizability of the model. We also implemented the qualitative replication approach, in which we train the SCCA model independently in Generation R and compare the results across the two cohorts. Note that the sample size in ABCD is an example from one train-test split

(2). Various parts are worded a little strongly. For example, “in order to safeguard against overfitting ...”. PCA on the connectivity matrices alone would not safeguard against overfitting. It would reduce the dimension of the data, which can help minimize overfitting. In general, it would be good if the authors could be very specific with their language. That will help any readers not misunderstand their findings.

(2). We thank the Reviewer for the comments. We have revised the manuscript accordingly in several locations, see below some examples:

Results (page 7):

“Importantly, the analyses in ABCD_{Training} sets and ABCD_{Test} sets were fully separated to help minimize the potential for data leakage (**Figure 1**).”

Results (page 3):

“To help mitigate possible overfitting problems, the connectivity matrices underwent dimensionality reduction by principal component analysis (PCA) with a weighting scheme (see Methods).”

Discussion (page 13):

“To improve the robustness and generalizability of brain-behavior associations in a fully independent sample, which is largely absent or sub-optimally done in previous research in doubly multivariate psychiatric neuroimaging literature.”

“Using the largest multicohort study investigating the multivariate brain-behavior associations in pre-adolescence, the enhanced statistical power allowed us to examine whether robust associations can be detected and generalized in the general population.”

(3). More than 10 splits might be helpful. There is a large variation in results for the first canonical factor ($r=0.09-0.17$).

(3). We thank the Reviewer for this suggestion. We agree that additional splits would be helpful when inspecting variations resulting from sampling. Thus, we resampled 30 splits in ABCD. Briefly, the results with 30 splits were similar to the results from the 10 splits, showing a similar degree of generalizability in Generation R. We chose 30 splits because of our considerations regarding the computational demands and based on the central limit theorem.

The new results were shown in **Figure 2-5** and **Supplementary Table 3**, as well as the Results section. Please see the detailed results above in our reply to **Question #2** from **Reviewer #1**.

(4). The culture between the Netherlands and the US are very different. The authors hint at this, but I think they could do a better job of situating this in terms of generalization. For example, even within ABCD prediction performance across different demographic groups exist, suggesting that generalizing mental health measures across cultures is hard. Indeed, we know that different communities have differences in symptom presentation, as well as differing perspectives on mental health more broadly. In all it is not surprising that models might not generalize across cultures. In other words, a lack of generalization might not be due to overfitting, bias, or poor methods. It may in

fact represent a true difference in brain behavior associations between cohorts. See for example Tejavibulya L, et al. Predicting the future of neuroimaging predictive models in mental health. *Mol Psychiatry*. 2022 Aug;27(8):3129-3137. doi: 10.1038/s41380-022-01635-2. Epub 2022 Jun 13. PMID: 35697759; PMCID: PMC9708554.

This is especially true given the small effect sizes in the study. A correlation of $<.1$ does not have much room to lose any explained variance. For example, if the effects were around $r=.5$, losing half of the effect would still lead to a significant result. But with $r=.1$ losing half of the effect is likely to be insignificant.

(4). We thank the Reviewer for this key comment. We agree that brain-behavior associations could be different across cultures and the associations are usually intertwined with other factors. This is now discussed in the Discussion section.

Discussion (page 16):

“Another possible explanation is that brain-behavior associations differ across populations and cultures due to unconsidered confounders and differences in presentation of symptoms. Model failures are usually interweaved with other factors³², such as differences in reporting preference and symptom presentation in diverse populations, which may correspond to divergent neurobiological underpinnings. The internally valid associations in ABCD could be cohort-specific effects that are not entirely consistent with Generation R. Although the 8-syndrome structure of CBCL was shown to be stable across different societies^{33,34}, our results could reflect, to some extent, the different brain-symptom construction across cultures.”

While acknowledging the cultural differences, we believe that the ultimate aim of prediction models is to identify brain biomarkers that are clinically useful, not only for one cultural group. Therefore, cultural differences could be considered when training the model to improve generalizability. We discussed this further in the Discussion section.

Discussion (page 15):

“Nevertheless, the primary goal of machine learning models is to identify brain biomarkers that can improve the diagnoses, treatment, and prevention of psychiatric disorders, not only for one specific group. A more ideal approach could be similar to the risk calculator developed in medical research³⁵, a more standardized protocol in genetic association studies³⁶, or some prediction pipelines developed in non-psychiatric studies⁷. The common characteristic of these examples is that model weights (“gold-standard”) are applied to different populations with diverse backgrounds, which has a high demand for the model as well as the identification of *common biomarkers*. If model performance varied across some groups (e.g., sex, age, cultural backgrounds), the important predictors could be included in the extension and further validation of models to create a more generalizable, clinically useful model.”

(5). Relatedly, the authors make the point that different results might be seen in clinical samples and that these studies largely draw from healthy individuals. This is an important point as the authors also note that for clinical psychiatric care, we would a biomarker that generalizes well. But the authors do not test in a clinical group. So some of the writing about how these results impact mental health research might be a bit over interpreted.

We thank the Reviewer for this feedback. We acknowledge that the implication of our population-based approach might not be extended to clinical studies. We updated the manuscript and emphasized that our study was especially informative for the general population and modified our Discussion accordingly.

Discussion (page 13):

“While these results reinforce previous work demonstrating the potential for brain-based dimensions of psychiatric problems, they also highlight the problem of the generalizability of findings in psychiatric neuroimaging studies, especially in the general population.”

“In our study, the lack of this ‘gold-standard’ generalizability in an external, independent sample suggests limited external validity, meaning that the dimensions cannot be applied to other datasets as a potential biomarker in the general population.”

Discussion (page 17):

“Second, the conclusions drawn from the current study might not generalize to clinical groups. Although similar poor out-of-sample multivariate associations were seen in clinical samples³⁷, prediction models built in clinical studies might be more robust due to potentially larger effect sizes. Yet, biomarkers emerging from the general population are useful in screening high-risk individuals, prevention, and health education, which are also of pivotal importance for improving health care quality.”

(6). Did the authors try training in Generation R and testing in ABCD? It would be worth knowing if the same pattern of generalization (or lack thereof) is observed.

(6). We chose to use ABCD as the discovery set for two main reasons. First, multivariate methods, especially CCA, are highly prone to overfitting when the sample/feature ratio is small³⁸. To mitigate the possibility of overfitting, we need a large data set. With the sample size of ABCD (n=4,892) and Generation R (n=2,034), it is safer to train the model in the larger sample. Second, ABCD is a multi-site cohort with 21 study sites, which makes it a better sample to do leave-sites-out validation, which is a better approximation for true out-of-sample settings.

However, we find this to be an important point, and following the suggestions from the Reviewer, we tried to train SCCA in Generation R and test the out-of-study performance in ABCD. We ran the same analysis pipe with GenR as the discovery set and ABCD as the external validation set. Overall, no associations were generalizable to ABCD. This is expected as the sample size of GenR is likely not sufficient to capture robust multivariate associations with this particular dimensionality of brain features.

Table*Canonical correlations across Generation R and ABCD*

	Generation R (n = 2,043)					ABCD n = 4,892
	Canonical Correlations	Training (n=1,430)	Test (n=613)	Sparsity		
Split 1	r_1	0.39	0.08	rs-fMRI	0.9	0.01
	r_2	0.33	0.08	CBCL	0.5	0.008
	r_3	0.36	0.08			0.02
Split 2	r_1	0.42	0.03	rs-fMRI	0.7	0.003
	r_2	0.27	0.03	CBCL	0.8	0.03
	r_3	0.29	0.04			0.04
Split 3	r_1	0.23	0.02	rs-fMRI	0.3	0.02
	r_2	0.20	0.05	CBCL	0.9	0.007
	r_3	0.16	0.06			0.02
Split 4	r_1	0.36	0.07	rs-fMRI	0.5	0.006
	r_2	0.36	0.05	CBCL	0.7	0.003
	r_3	0.22	0.04			0.04
Split 5	r_1	0.13	0.01	rs-fMRI	0.1	0.02
	r_2	0.12	0.04	CBCL	0.5	0.01
	r_3	0.13	0.05			0.01
Split 6	r_1	0.32	0.05	rs-fMRI	0.4	0.008
	r_2	0.19	0.03	CBCL	0.9	0.001
	r_3	0.17	0.03			0.002
Split 7	r_1	0.33	0.05	rs-fMRI	0.5	0.004
	r_2	0.16	0.12*	CBCL	0.9	0.07
	r_3	0.19	0.02			0.04
Split 8	r_1	0.10	0.01	rs-fMRI	0.1	0.001
	r_2	0.10	0.01	CBCL	0.1	0.02
	r_3	0.15	0.02			0.01
Split 9	r_1	0.29	0.02	rs-fMRI	0.3	0.007
	r_2	0.24	0.01	CBCL	0.4	0.03
	r_3	0.27	0.03			0.03
Split 10	r_1	0.37	0.11*	rs-fMRI	0.8	0.02
	r_2	0.38	0.01	CBCL	0.5	0.005
	r_3	0.37	0.08			0.008

(7). It would also be good to explicitly define terms like generalization and replication. While the terms are often used interchangeably. They can be different things to different researchers.

(7). We thank the Reviewer for this crucial question and have noticed this even in the literature. We have revised the Methods section accordingly.

Methods (page 28):

“We utilized two approaches to test the external validity: the qualitative replication and a gold-standard generalizability test. Here, we define replication as repeating the analyses in different settings and observing the similarities of findings across studies, while generalization refers to the same statistical model successfully making predictions in different populations. Replication provides evidence of important correlations, and further generalizability tests are conducted to provide a realistic possibility for extending these discoveries into clinical applications.”

Reviewer #3

In this study the authors assess the external cross-validation of data-driven child psychopathology from rsfMRI connectivity and CBCL by utilizing the ABCD study baseline and the Generation R study samples. The question is important and merits scrutiny and publications from different groups, even if other work has already alluded to this issue.

We thank the Reviewer for the valuable comments and suggestions.

(1). The study’s scope is presented in a manner that utilizing only one method does not seem comprehensive enough and other common methods than SCCA should be assessed and included somewhere, potentially in the supplementary section. Maybe at least one from the Kernel family (kernel ridge regression or support vector regression), as well as connectome-based predictive modeling, since these are used widely. Currently, only small variations in the SCCA are shown in the supplementary section.

(1). We thank the Reviewer for the advice. We clarified in our Introduction that our study focuses on **“doubly” multivariate techniques** (“many-to-many” associations), in which multiple brain features and multiple behavioral features are fed into the model simultaneously, and the latent spaces from two datasets are captured. The **CCA and PLS (Partial Least Squares)** family emerged as two of the most widely used doubly multivariate methods and have been increasingly applied in neuroimaging studies. Other types of multivariate techniques, such as the kernel family and connectome-based predictive modeling the Reviewer mentioned, usually involve many brain features and one behavior feature (e.g., using a linear model to predict one cognitive ability or presence/absence of one disorder). Although similar problems of out-of-sample performance were reported²³, some standardized prediction protocols have been established for these many-to-one prediction models⁷. Further, given the already extensive reporting required to address the main research question of the current study (i.e., proper generalizability testing a doubly multivariate technique), we have decided adding new models is beyond the scope of the current study. We have tried to clarify this point in the Introduction, particularly in making clear the goal of the present study:

Introduction (page 4):

“Multivariate studies have either adopted multiple-to-one approach (e.g., support vector machine family) using many brain features to predict cognition or diagnoses of disease, or multiple-to-multiple (doubly) methods that can assess the covariation of many neural phenotypes (e.g., brain activity across regions) and many behavioral features simultaneously. One widely used doubly multivariate method in neuroimaging is canonical correlation analysis (CCA), a technique that aims to identify the common variation across phenotypes and dissect their complex relationships into a small number of distinct components⁴⁰.”

(2). The correlations are not particularly high even in the internal cross-validation (e.g. $r = .13$ for the primary LV), which is consistent with other ABCD-based studies showing that, other than the cognitive domain, measures in the personality and mental health domains have low brain-phenotype associations (e.g. Chen, J., Tam, A., Kebets, V. et al. Shared and unique brain network features predict cognitive, personality, and mental health scores in the ABCD study. Nat Commun 13, 2217 (2022).). Given this, it is expected that such already small out-of-sample r would not survive an external cross-validation. I am wondering what the upper-bound of the cross-dataset validation is based on just the reliability of the psychopathology latent variables between ABCD and Gen R (i.e., ignoring brain)?

(2). We agree that the effect sizes of brain-symptoms associations are much lower than those with cognition, which makes it challenging to identify generalizable associations across populations. To address this question, we used data only from the GenR study. Specifically, we used the CBCL data in GenR to a.) calculate the CBCL canonical variates by training the SCCA in GenR and b.) deriving the canonical variates in GenR by apply the weight vectors obtained from training the SCCA in ABCD. We then calculated the Pearson correlation between these two sets of canonical variates (i.e., trained in GenR vs trained in ABCD). The correlation for the latent score of attention problems was $r = 0.90$, for rule-breaking behaviors is $r = 0.93$, and for anxious/depressed is $r = 0.87$. Thus, even when training in two separate samples, the results of canonical variate loadings are highly similar.

Further, within ABCD_{Training}, we repeatedly subsampled the data and fit the SCCA. Across the repeated subsamples, the canonical loadings for the CBCL canonical variates were relatively stable (**Figure 4a**). Thus, we suggest that at least the CBCL latent variables are relatively reliable. In fact, the high correlations of the CBCL canonical variate scores across cohorts could be expected according to the similar CBCL loadings in the qualitative replication (**Figure 5**), suggesting brain connectivity difference is underlying the suboptimal generalizability, not the psychopathology phenotypes. See also above our reply to **Reviewer #1 Comment #6b** where we show the divergence of the brain phenotypes is likely driving the reduction in generalizability across the samples. This has been addressed in the Results and Discussion sections:

Results (page 9):

“While relatively stable contribution from the CBCL syndrome scores was observed, the instability of rs-fMRI canonical loadings manifested through more variability and less clear patterns in the canonical loadings for brain PCs (**Figure 4a, 4b**).”

Discussion (page 16):

“Third, resting-state fMRI data has intrinsic high inter-individual variability and smaller effect sizes at the individual level than other brain measures in psychiatry, thus extracting clinically important signals on an individual basis is difficult and generalizability across cohorts could be

especially challenging. This can be seen from our results: the psychopathology profiles were relatively stable within ABCD as well as across cohorts, but the brain phenotypes associated with the behavioral profiles were highly unstable.”

(3). line 78: I would also cite previous work recommending these such as e.g.: Scheinost, D., Noble, S., Horien, C., Greene, A. S., Lake, E. M., Salehi, M., ... & Constable, R. T. (2019). Ten simple rules for predictive modeling of individual differences in neuroimaging. *NeuroImage*, 193, 35-45). Additionally, I would include some studies from non-psychiatric neuroimaging domain that involve external cross-validation (in addition to internal) after “medical research” and before the “psychiatric neuroimaging” to both temper the sentence and make the gap clearer (e.g. Avery, E. W., Yoo, K., Rosenberg, M. D., Greene, A. S., Gao, S., Na, D. L., ... & Chun, M. M. (2020). Distributed patterns of functional connectivity predict working memory performance in novel healthy and memory-impaired individuals. *Journal of cognitive neuroscience*, 32(2), 241-255.; Kardan, O., Stier, A. J., Cardenas-Iniguez, C., Schertz, K. E., Pruin, J. C., Deng, Y., ... & Rosenberg, M. D. (2022). Differences in the functional brain architecture of sustained attention and working memory in youth and adults. *Plos Biology*, 20(12), e3001938.

(3). We thank the Reviewer for the suggestions and included the recommendations and studies on non-psychiatric neuroimaging prediction models in the Introduction.

Introduction (page 5):

“This has been widely implemented in the validation of prediction models in medical research^{4,5} and recommended as a necessary step in prediction models⁶. While several non-psychiatric neuroimaging studies have established more standardized analysis pipelines⁷⁻⁹, most multivariate psychiatric neuroimaging studies have not generally adopted these stringent external validation strategies¹⁰⁻¹⁶.”

(4). line 97: The sample size from ABCD in this study is much larger than other studies with adequate exclusion of head motion using ABCD rsfMRI (e.g. Wang, Z., Zhou, X., Gui, Y. et al. Multiple measurement analysis of resting-state fMRI for ADHD classification in adolescent brain from the ABCD study. *Transl Psychiatry* 13, 45 (2023). <https://doi.org/10.1038/s41398-023-02309-5> or Sripada, C., Rutherford, S., Angstadt, M. et al. Prediction of neurocognition in youth from resting state fMRI. *Mol Psychiatry* 25, 3413–3421 (2020). <https://doi.org/10.1038/s41380-019-0481-6>). Please elaborate on this in the methods or discussion.

(4). We thank the Reviewer for the comment. We extracted the raw resting-state time series data from the recently released ABCD-BIDS Community Collection (ABCC), which is a community-shared and continually updated ABCD neuroimaging dataset. To address the concern of sample size, we implemented a stricter quality control process in the new data set we included. The final sample size ($n = 4,892$) was similar to studies using the earlier released resting-state data from the ABCC collection (Marek, S. et al. (2022), $n = 3,928$)⁴¹. The detailed inclusion criteria were updated in the Methods section.

Methods (page 19):

“Of the 9,441 children whose rs-fMRI data were available, we excluded 3,720 children who failed the quality control of the resting-state connectivity data (see below), 220 children with incidental findings, and 14 children with any missingness in behavioral measures and covariates.

For families with multiple participants, one twin or sibling was randomly included (595 excluded). Accordingly, data from 4,892 participants, of which around 6.9% had clinically relevant total problem symptom scores, were available for analysis in ABCD.”

Methods (page 21):

“Participants were excluded based on the ABCD recommended guidelines (imgincl_rsfmri_include = 1), which involve raw and postprocessing quality control, passed FreeSurfer QC, had more than 375 rs-fMRI frames after censoring, and other cut-off scores (see ABCD Recommended Imaging Inclusion), 1,310 participants were excluded due poor quality. We additionally excluded 2,410 participants with excessive motion (mean framewise displacement (FD) higher than 0.25 mm)⁴², and 220 participants with clinically relevant incidental findings.”

References

1. Insel, T. *et al.* Research Domain Criteria (RDoC): Toward a New Classification Framework for Research on Mental Disorders. *Am. J. Psychiatry* **167**, 748–751 (2010).
2. Krueger, R. F. & Bezdjian, S. Enhancing research and treatment of mental disorders with dimensional concepts: toward DSM-V and ICD-11. *World Psychiatry* **8**, 3–6 (2009).
3. Hudziak, J. J., Achenbach, T. M., Althoff, R. R. & Pine, D. S. A dimensional approach to developmental psychopathology. *Int. J. Methods Psychiatr. Res.* **16**, S16–S23 (2007).
4. Siontis, G. C. M., Tzoulaki, I., Castaldi, P. J. & Ioannidis, J. P. A. External validation of new risk prediction models is infrequent and reveals worse prognostic discrimination. *J. Clin. Epidemiol.* **68**, 25–34 (2015).
5. Carrión, R. E. *et al.* Personalized Prediction of Psychosis: External Validation of the NAPLS-2 Psychosis Risk Calculator With the EDIPPP Project. *Am. J. Psychiatry* **173**, 989–996 (2016).
6. Scheinost, D. *et al.* Ten simple rules for predictive modeling of individual differences in neuroimaging. *NeuroImage* **193**, 35–45 (2019).
7. Shen, X. *et al.* Using connectome-based predictive modeling to predict individual behavior from brain connectivity. *Nat. Protoc.* **12**, 506–518 (2017).
8. Avery, E. W. *et al.* Distributed Patterns of Functional Connectivity Predict Working Memory Performance in Novel Healthy and Memory-impaired Individuals. *J. Cogn. Neurosci.* **32**, 241–255 (2020).
9. Kardan, O. *et al.* Differences in the functional brain architecture of sustained attention and working memory in youth and adults. *PLOS Biol.* **20**, e3001938 (2022).
10. Buch, A. M. Molecular and network-level mechanisms explaining individual differences in autism spectrum disorder. *Nat. Neurosci.* **26**, (2023).
11. Xia, C. H. *et al.* Linked dimensions of psychopathology and connectivity in functional brain networks. *Nat. Commun.* **9**, 3003 (2018).
12. Linke, J. O. *et al.* Shared and Anxiety-Specific Pediatric Psychopathology Dimensions Manifest Distributed Neural Correlates. *Biol. Psychiatry* **89**, 579–587 (2021).
13. Moser, D. A. *et al.* Multivariate Associations Among Behavioral, Clinical, and Multimodal Imaging Phenotypes in Patients With Psychosis. *JAMA Psychiatry* **75**, 386 (2018).
14. Alnæs, D., Kaufmann, T., Marquand, A. F., Smith, S. M. & Westlye, L. T. Patterns of sociocognitive stratification and perinatal risk in the child brain. *Proc. Natl. Acad. Sci.* **117**, 12419–12427 (2020).
15. Tozzi, L., Tuzhilina, E., Glasser, M. F., Hastie, T. J. & Williams, L. M. Relating whole-brain functional connectivity to self-reported negative emotion in a large sample of young adults using group regularized canonical correlation analysis. *NeuroImage* **237**, 118137 (2021).
16. Xiao, X. *et al.* Brain Functional Connectome Defines a Transdiagnostic Dimension Shared by Cognitive Function and Psychopathology in Preadolescents. *Biol. Psychiatry* S0006322323015937 (2023) doi:10.1016/j.biopsych.2023.08.028.
17. Esteban, O. *et al.* fMRIPrep: a robust preprocessing pipeline for functional MRI. *Nat. Methods* **16**, 111–116 (2019).
18. Gordon, E. M. *et al.* Generation and Evaluation of a Cortical Area Parcellation from Resting-State Correlations. *Cereb. Cortex* **26**, 288–303 (2016).
19. Fischl, B. *et al.* Whole Brain Segmentation. *Neuron* **33**, 341–355 (2002).
20. Sripada, C. *et al.* Prediction of Neurocognitive Profiles in Youth From Resting State fMRI. (2018) doi:10.1101/495267.
21. Lauritzen, T. Z., D’Esposito, M., Heeger, D. J. & Silver, M. A. Top-down flow of visual spatial attention signals from parietal to occipital cortex. *J. Vis.* **9**, 18–18 (2009).

22. Smith, S. M. *et al.* A positive-negative mode of population covariation links brain connectivity, demographics and behavior. *Nat. Neurosci.* **18**, 1565–1567 (2015).
23. Tian, Y. & Zalesky, A. Machine learning prediction of cognition from functional connectivity: Are feature weights reliable? *NeuroImage* **245**, 118648 (2021).
24. Mihalik, A. *et al.* Multiple Holdouts With Stability: Improving the Generalizability of Machine Learning Analyses of Brain–Behavior Relationships. *Biol. Psychiatry* **87**, 368–376 (2020).
25. Liu, T. T. Noise contributions to the fMRI signal: An overview. *NeuroImage* **143**, 141–151 (2016).
26. Achenbach, T.M. & Rescorla, L. A. *Manual for the ASEBA School-Age Forms & Profiles.* (University of Vermont, Research center for children, youth, & families., 2001).
27. Thurber, S. Note on Truncated T Scores in Discrepancy Studies with the Child Behavior Checklist and Youth Self Report. *Arch. Assess. Psychol.* **2**, 73–80 (2012).
28. Drysdale, A. T. *et al.* Resting-state connectivity biomarkers define neurophysiological subtypes of depression. *Nat. Med.* **23**, 28–38 (2017).
29. Witten, D. M., Tibshirani, R. & Hastie, T. A penalized matrix decomposition, with applications to sparse principal components and canonical correlation analysis. *Biostatistics* **10**, 515–534 (2009).
30. Knief, U. & Forstmeier, W. Violating the normality assumption may be the lesser of two evils. *Behav. Res. Methods* **53**, 2576–2590 (2021).
31. Voldsbekk, I. *et al.* Delineating disorder-general and disorder-specific dimensions of psychopathology from functional brain networks in a developmental clinical sample. *Dev. Cogn. Neurosci.* **62**, 101271 (2023).
32. Greene, A. S. *et al.* Brain–phenotype models fail for individuals who defy sample stereotypes. *Nature* **609**, 109–118 (2022).
33. Rescorla, L. A., Althoff, R. R., Ivanova, M. Y. & Achenbach, T. M. Effects of society and culture on parents’ ratings of children’s mental health problems in 45 societies. *Eur. Child Adolesc. Psychiatry* **28**, 1107–1115 (2019).
34. Ivanova, M. Y. *et al.* Testing the 8-Syndrome Structure of the Child Behavior Checklist in 30 Societies. *J. Clin. Child Adolesc. Psychol.* **36**, 405–417 (2007).
35. Collins, G. S. & Altman, D. G. An independent external validation and evaluation of QRISK cardiovascular risk prediction: a prospective open cohort study. *BMJ* **339**, b2584–b2584 (2009).
36. ADHD Working Group of the Psychiatric Genomics Consortium (PGC) *et al.* Discovery of the first genome-wide significant risk loci for attention deficit/hyperactivity disorder. *Nat. Genet.* **51**, 63–75 (2019).
37. Ji, J. L. *et al.* Mapping brain-behavior space relationships along the psychosis spectrum. *eLife* **10**, e66968 (2021).
38. Helmer, M. *et al.* *On stability of Canonical Correlation Analysis and Partial Least Squares with application to brain-behavior associations.*
<http://biorxiv.org/lookup/doi/10.1101/2020.08.25.265546> (2020) doi:10.1101/2020.08.25.265546.
39. Lerma-Usabiaga, G., Mukherjee, P., Ren, Z., Perry, M. L. & Wandell, B. A. Replication and generalization in applied neuroimaging. *NeuroImage* **202**, 116048 (2019).
40. Wang, H.-T. *et al.* Finding the needle in a high-dimensional haystack: Canonical correlation analysis for neuroscientists. *NeuroImage* **216**, 116745 (2020).
41. Marek, S. *et al.* Reproducible brain-wide association studies require thousands of individuals. *Nature* **603**, 654–660 (2022).
42. Parkes, L., Fulcher, B., Yücel, M. & Fornito, A. An evaluation of the efficacy, reliability, and sensitivity of motion correction strategies for resting-state functional MRI. *NeuroImage* **171**, 415–436 (2018).

4th Jan 24

Dear Professor Tiemeier,

Your manuscript titled "Multivariate brain-based dimensions of child psychiatric problems: degrees of generalizability" has now been seen by our reviewers, whose comments appear below. In light of their advice I am delighted to say that we are happy, in principle, to publish a suitably revised version in *Communications Psychology* under the open access CC BY license (Creative Commons Attribution v4.0 International License).

We therefore invite you to revise your paper one last time to address the remaining concerns of our reviewers and a list of editorial requests. At the same time we ask that you edit your manuscript to comply with our format requirements and to maximise the accessibility and therefore the impact of your work.

EDITORIAL REQUESTS:

SUBMISSION INFORMATION:

OPEN ACCESS:

Communications Psychology is a fully open access journal. Articles are made freely accessible on publication under a [CC BY license](http://creativecommons.org/licenses/by/4.0) (Creative Commons Attribution 4.0 International License). This license allows maximum dissemination and re-use of open access materials and is preferred by many research funding bodies.

For further information about article processing charges, open access funding, and advice and support from Nature Research, please visit <https://www.nature.com/commspsychol/article-processing-charges>

At acceptance, you will be provided with instructions for completing this CC BY license on behalf of all authors. This grants us the necessary permissions to publish your paper. Additionally, you will be asked to declare that all required third party permissions have been obtained, and to provide billing information in order to pay the article-processing charge (APC).

* **DATA AVAILABILITY:**

[link redacted]

Best regards,

Antonia Eisenkoeck

Antonia Eisenkoeck
Senior Editor
Communications Psychology

REVIEWERS' COMMENTS:

Reviewer #1 (Remarks to the Author):

The authors have responded thoroughly to my previous concerns, and the revised manuscript has been strengthened considerably. I have no further requests for changes to the manuscript.

Reviewer #2 (Remarks to the Author):

The authors have addressed my comments.

Reviewer #3 (Remarks to the Author):

All of my main concerns have been addressed in the revised manuscript. I have no further comments.

Reviewer #4 (Remarks to the Author):

I am happy with the revisions made.

[Editorial note: Reviewer #4 co-reviewed with Reviewer #1, which is why they did not show up in the first round of reviews.]